# Optimized PAR-2 RING dimerization mediates cooperative and selective membrane binding for robust cell polarity

Tom Bland [ID][1,2], Nisha Hirani[1], David C Briggs [ID][1], Riccardo Rossetto [ID][3], KangBo Ng [ID][1,2], Ian A Taylor [ID][1], Neil Q McDonald [ID][1,4], David Zwicker [ID][3] & Nathan W Goehring [ID][1,2]✉

## Abstract

**Cell polarity networks are defined by quantitative features of their constituent feedback circuits, which must be tuned to enable robust and stable polarization, while also ensuring that networks remain responsive to dynamically changing cellular states and/or spatial cues during development. Using the PAR polarity network as a model, we demonstrate that these features are enabled by the dimerization of the polarity protein PAR-2 via its N-terminal RING domain. Combining theory and experiment, we show that dimer affinity is optimized to achieve dynamic, selective, and cooperative binding of PAR-2 to the plasma membrane during polarization. Reducing dimerization compromises positive feedback and robustness of polarization. Conversely, enhanced dimerization renders the network less responsive due to kinetic trapping of PAR-2 on internal membranes and reduced sensitivity of PAR-2 to the anterior polarity kinase, aPKC/PKC-3. Thus, our data reveal a key role for a dynamically oligomeric RING domain in optimizing interaction affinities to support a robust and responsive cell polarity network, and highlight how optimization of oligomerization kinetics can serve as a strategy for dynamic and cooperative intracellular targeting.**

**Keywords** PAR Proteins; Cell Polarity; RING Domain Dimerization Self-organization; *C. elegans* Zygote
**Subject Category** Cell Adhesion, Polarity & Cytoskeleton

## Introduction

Robust polarization of cells typically relies on feedback pathways to amplify and stabilize molecular asymmetries (Chau et al, 2012; Gierer and Meinhardt, 1972; Meinhardt and Gierer, 1974; Mogilner et al, 2012; Wedlich-Soldner et al, 2003). Critically, this feedback must be appropriately configured to balance key tradeoffs in the potential behaviors of a system. For example, increased feedback may render a system more sensitive to polarizing cues and enhance the stability of the resulting polarized state, but this may come at the cost of either responding to inappropriate cues such as random fluctuations or failing to adapt to signals that change in space and time (Jilkine and Edelstein-Keshet, 2011). While feedback is clearly implicated in the intracellular patterning mechanisms that underlie cell polarity, it is often difficult to obtain direct and quantitative measures of feedback in living systems, let alone be able to directly link feedback behavior to specific molecular activities such that feedback can be manipulated to test its effects on system behavior. This is due in part to the inherent complexity and redundancy of polarity networks that make it difficult to isolate core feedback circuits. It is also technically challenging to perform the required dose–response measurements in vivo with sufficient precision and accuracy (Graziano et al, 2017). Thus, in many cases, we lack rigorous quantitative assessment of what are often purported to be core pattern-forming features of cell polarity networks.

The PAR (partitioning defective) polarity network is one such example. At the core of the PAR polarity network is a set of cross-inhibitory interactions that result in mutually exclusive localizations of distinct groups of peripherally associated PAR proteins on the plasma membrane (Goehring, 2014; Lang and Munro, 2017). In the *C. elegans* zygote, polarization is induced by the centrosome, which induces cortical actomyosin flows that segregate one group of PAR proteins, the so-called aPARs that include PAR-3, PAR-6, and PKC-3 (aPKC), into an anterior membrane-associated domain (Cowan and Hyman, 2004; Goehring et al, 2011b; Munro et al, 2004; Zhao et al, 2019) (Fig. 1A). The anterior polarity kinase PKC-3 phosphorylates a second group of posterior polarity proteins (pPARs) that include PAR-1, PAR-2, and LGL-1 to restrict their localization (Beatty et al, 2010; Betschinger et al, 2003; Hao et al, 2006; Hoege et al, 2010; Hurov et al, 2004; Plant et al, 2003). Prior to symmetry-breaking, pPAR proteins are initially depleted from the plasma membrane by aPARs and load onto the posterior as aPARs are segregated into the anterior (Cuenca et al, 2003; Hurov et al, 2004; Tabuse et al, 1998). The posterior polarity kinase PAR-1 in turn targets PAR-3, helping to restrict its localization to the anterior plasma membrane (Benton and St Johnston, 2003; Guo

[1]Francis Crick Institute, London NW1 1AT, UK. [2]Institute for the Physics of Living Systems, University College London, London, UK. [3]Max Planck Institute for Dynamics and Self-Organization, Göttingen, Germany. [4]Institute of Structural and Molecular Biology, Department of Biological Sciences, Birkbeck College, London WC1E 7HX, UK. ✉E-mail: nate.goehring@crick.ac.uk

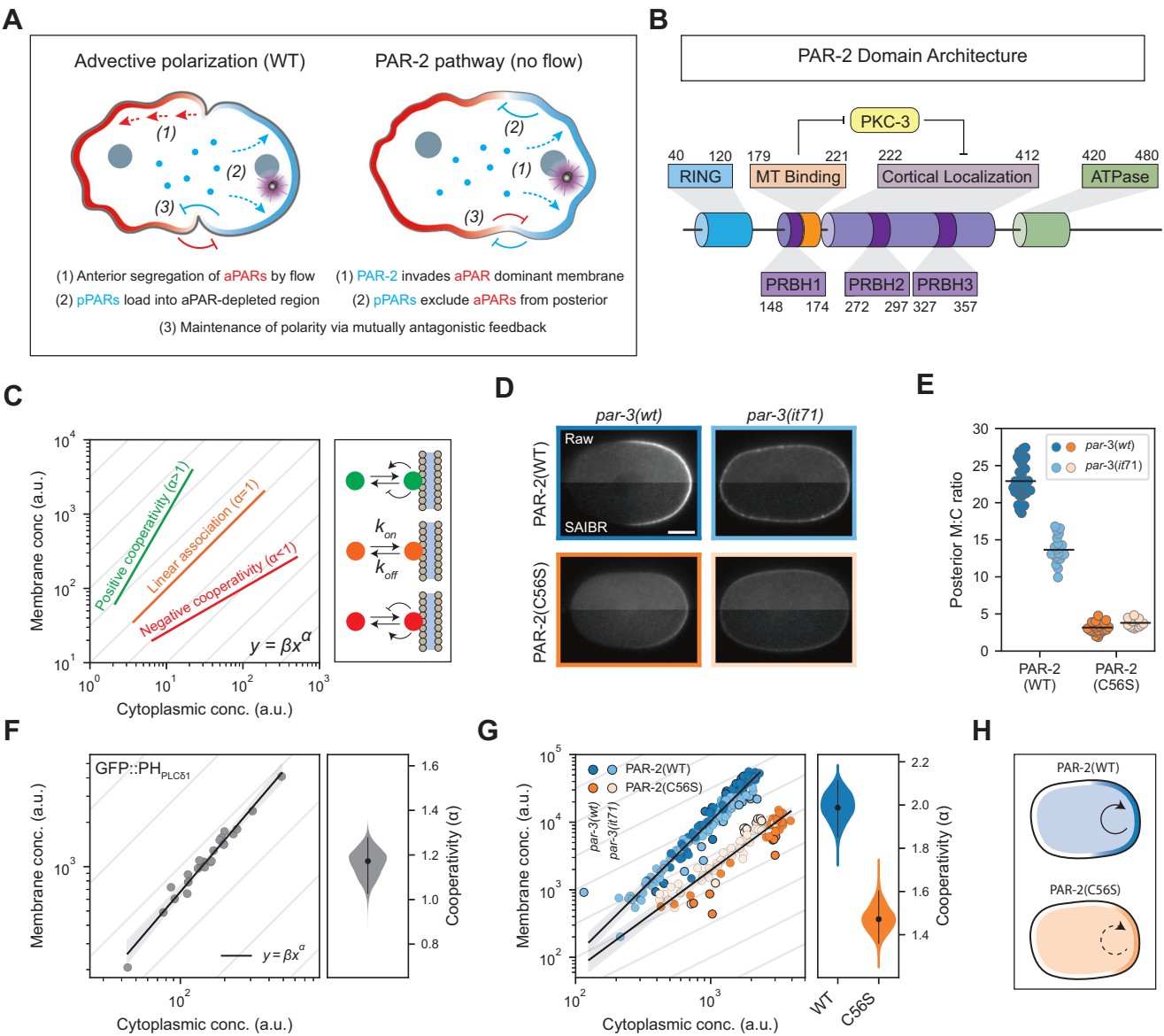

**Figure 1. The PAR-2 RING domain drives cooperative membrane association.**

(A) Schematic of PAR polarization via advective polarization (WT) and the PAR-2 pathway (no flow). Note the change in the ordering of PAR segregation events. In normal conditions, aPARs are polarized first by flows and pPARs then load into the posterior as aPARs are cleared. By contrast, in the PAR-2 pathway, local loading of PAR-2 initiates polarization, creating a pPAR domain, which then excludes aPARs from the posterior. (B) Schematic of PAR-2 functional domains. (C) Cooperative membrane association causes a deviation from linearity in the mapping between cytoplasmic and membrane concentrations. A molecule that binds and unbinds from the membrane with linear kinetics (i.e., rates independent of concentrations) will have a linear relationship between cytoplasmic and membrane concentrations, equivalent to a slope of 1 on a log–log plot (orange). Cooperative membrane association, in which the rates of membrane association and/or dissociation change as a function of membrane concentration, leads to a nonlinear mapping between cytoplasmic and membrane concentrations, equivalent to a slope >1 on a log–log plot in the case of positive cooperativity (green) or <1 in the case of negative cooperativity (red) (see "Scoring cooperativity"). (D) Raw and SAIBR-corrected images of mNG::PAR-2(WT) and mNG::PAR-2(C56S) in polarized (*par-3(WT)*) and uniform (*par-3(it71)*) conditions. (E) Quantification of the posterior membrane to cytoplasmic ratio for the conditions in (D). (F) Quantification of membrane and cytoplasmic GFP::PH$_{PLC\delta1}$ concentrations in embryos with varying total amounts of GFP::PH$_{PLC\delta1}$ (achieved by RNAi). Black line shows linear fit to log-transformed data and 95% confidence band. Right: probability distribution of the cooperativity score (slope of the linear fit), calculated by bootstrapping. A cooperativity score close to 1 reveals near-linear membrane association. (G) Quantification of membrane and cytoplasmic PAR-2 concentrations in embryos with varying total amounts of PAR-2. Data from both polarized cells (dark points) and uniform cells (light points) are pooled, for both wild-type PAR-2 (blue) and RING-mutant (C56S) PAR-2 (orange). Membrane concentration measurements are limited to the posterior-most 20% of the cell. Black lines show linear fits to log-transformed data and 95% confidence bands. Right: probability distributions of the cooperativity scores for PAR-2(WT) and PAR-2(C56S). (H) Schematic of PAR-2 cooperative membrane association. Posterior membrane concentrations of wild-type PAR-2 are amplified by cooperativity. Cooperativity is diminished in RING-mutant (C56S) cells, resulting in reduced membrane concentrations. Data Information: In (D), scale bar = 10 μm. (E–G) Datapoints represent individual embryos. (E) Mean indicated. (F, G) Best fits to the full dataset (dots) are shown with probability distributions of cooperativity (violin plot) and 95% confidence intervals (lines) calculated by bootstrapping. Additional statistics are available in Table EV2. Source data are available online for this figure.

and Kemphues, 1995; Motegi et al, 2011). Polarity is further re-enforced by an additional reciprocal cross-inhibitory circuit involving active CDC-42/PKC-3 in the anterior and the CDC-42 GAP, CHIN-1, in the posterior (Kumfer et al, 2010; Sailer et al, 2015).

At the same time, it is increasingly thought that simple cross-inhibitory reactions are insufficient to fully account for the behavior of the PAR network, most notably because pattern formation by the PAR network, like many other patterning networks, is thought to depend on nonlinear or bistable reaction dynamics that allow the system to support opposing membrane domains in distinct states (Arata et al, 2016; Dawes and Munro, 2011; Goehring et al, 2011b; Jilkine and Edelstein-Keshet, 2011; Lang and Munro, 2017; Meinhardt, 1982; Sailer et al, 2015). Indeed, quantitative dosage-phenotype maps reveal the nonlinear sensitivity of PAR network-dependent processes to perturbation (Rodrigues et al, 2023). How such nonlinearity arises in this system remains unclear. While a number of mechanisms have been postulated, including a potential role for oligomerization (Arata et al, 2016; Dawes and Munro, 2011; Goehring et al, 2011b; Lang et al, 2023; Sailer et al, 2015), direct measurements of nonlinear feedback are generally lacking. Thus, the key links between molecular activities, feedback responses, and network behavior remain poorly explored.

Here we focus on a subsystem of the PAR network centered on the posterior PAR protein PAR-2 (Fig. 1B). PAR-2 reversibly associates with the plasma membrane via a series of PRBH (**P**KC **R**esponsive **B**asic **H**ydrophobic) motifs that mediate electrostatic interaction with negatively charged lipids at the plasma membrane and its dissociation from the membrane is promoted by phosphorylation by PKC-3 (Hao et al, 2006; Motegi et al, 2011). PAR-2 is not believed to directly antagonize anterior PAR proteins, but rather supports polarity through what is known as the eponymous PAR-2 pathway (Ramanujam et al, 2018; Zonies et al, 2010). In this proposed pathway, binding of PAR-2 to centrosomal microtubules allows it to locally avoid phosphorylation by PKC-3 in the posterior at the time of polarization (Motegi et al, 2011). Once at the membrane, PAR-2 is thought to promote its own recruitment and becomes stabilized against the action of PKC-3 via its RING (**R**eally **I**nteresting **N**ew **G**ene) domain (Arata et al, 2016; Hao et al, 2006; Motegi et al, 2011). PAR-2 in turn recruits PAR-1 to the plasma membrane to support exclusion of PAR-3 from the posterior (Boyd et al, 1996; Motegi et al, 2011; Ramanujam et al, 2018).

Under normal conditions, the PAR-2 pathway acts to reinforce the initial polarization event in which aPARs are segregated into the anterior by cortical actomyosin flows (Fig. 1A, WT) (Hao et al, 2006). However, when actomyosin flows are absent and the initial segregation of aPARs fails, the PAR-2 pathway is sufficient to polarize the embryo in response to an actomyosin-independent centrosomal cue (Fig. 1A, no flow). The resulting posterior PAR-2 domain is then stable despite initially overlapping with aPARs (Goehring et al, 2011b; Motegi et al, 2011; Zonies et al, 2010). Once formed, this domain drives clearance of aPARs from the posterior to establish a properly polarized zygote (Motegi et al, 2011). While PAR-2 must be phosphorylated by PKC-3 to form a polarity domain (Hao et al, 2006; Hubatsch et al, 2019; Motegi et al, 2011; Ng et al, 2023) and thus is not capable of polarization in isolation, the ability of PAR-2 to invade and form a domain

within a uniform aPAR-dominated membrane suggests that PAR-2 may possess intrinsic self-amplifying feedback (Motegi et al, 2011). We therefore set out to identify and define the nature of this feedback, and quantitatively link it back to the molecular properties of PAR-2.

# Results

## PAR-2 exhibits RING domain-dependent positive feedback

As a first step, we sought to determine whether PAR-2 exhibits cooperative membrane binding. A simple model of reversible binding would be expected to yield a linear relationship between membrane and cytoplasmic concentrations with the membrane-to-cytoplasm (M:C) ratio given by $k_{on}/k_{off}$, where $k_{on}$ and $k_{off}$ define the respective membrane association and dissociation rate constants (Fig. 1C). By contrast, in systems with positive and/or negative cooperativity, M:C ratios will be concentration-dependent.

PAR-2 membrane concentrations appear visibly higher in polarized embryos in which PAR-2 is segregated within a posterior domain compared to aPAR-depleted embryos in which PAR-2 is uniform (Fig. 1D, Cuenca et al, 2003; Hao et al, 2006). Thus, simply measuring membrane:cytoplasmic (M:C) ratios for PAR-2 and PAR-2 variants under polarized (WT) and uniform (*par-3(it71)*) conditions should reveal whether membrane binding is dependent on membrane concentration. Specifically, if membrane binding is governed by mass action, M:C ratios should be constant between the two conditions despite the difference in concentration, at least when measured at the posterior where PKC-3 is absent.

To accurately measure M:C ratios, we combined autofluorescence correction via SAIBR (Rodrigues et al, 2022) with a machine-learning-based approach to assign local membrane and cytoplasmic fluorescence signals (see "Methods"). Strikingly, M:C ratios for PAR-2 were increased nearly twofold when PAR-2 was restricted to the posterior domain compared to when it was uniformly distributed, i.e. WT vs. *par-3(it71)* or PAR-2(S241A), in which the key PKC-3 phosphosite is mutated (Figs. 1D,E and EV1A) (Illukkumbura et al, 2023; Motegi et al, 2011). This observation argues against a simple mass action model for membrane binding for PAR-2.

We next sought to identify which features of PAR-2 were responsible for this apparent cooperativity. PAR-2 consists of an N-terminal RING domain, a region implicated in microtubule binding, a generally unstructured region enriched in basic–hydrophobic stretches that is required for membrane/cortex binding, and a C-terminal ATPase domain which appears dispensable for function (Fig. 1B) (Hao et al, 2006; Levitan et al, 1994; Motegi et al, 2011). Mutations affecting the ATPase domain or microtubule-binding regions have shown minimal effects on membrane localization under normal conditions (Hao et al, 2006; Motegi et al, 2011). By contrast, while the N-terminal domain of PAR-2 has been reported to be insufficient for membrane binding, variants of PAR-2 that either lack the RING domain or in which the RING is disrupted by mutation of a Zn-coordinating cysteine (C56S) exhibit reduced membrane binding (Hao et al, 2006). Thus, while the RING domain appears to lack intrinsic membrane

binding activity, it is required to potentiate membrane binding activity present elsewhere in the protein. We therefore introduced the C56S mutation into the *par-2* locus and measured the M:C ratio of PAR-2(C56S) in polarized and uniform conditions. In contrast to PAR-2(WT), PAR-2(C56S) exhibited M:C ratios that were similar between the segregated and uniform states (Fig. 1D,E; Table EV1). Thus the RING domain of PAR-2 appears to be important for the apparent cooperativity in PAR-2 membrane binding.

To explicitly measure the degree of positive feedback in PAR-2 membrane binding, we quantified the relationship between membrane and cytoplasmic concentrations in embryos subject to progressive reduction in total PAR-2 by RNAi. As a control, we examined embryos expressing a GFP fusion to the PIP$_2$-binding domain of PLCδ1 (GFP::PH$_{PLC\delta1}$) (Audhya et al, 2005; Hurley and Meyer, 2001), which we could progressively deplete by *gfp(RNAi)*. Fitting of the membrane to cytoplasmic concentrations with a phenomenological cooperative binding model yielded an effective exponent, α, of less than 1.2, consistent with minimal cooperativity (Fig. 1F, see "Methods"). By contrast, applying our method to embryos expressing endogenously tagged mNG::PAR-2 that were subject to progressive depletion of PAR-2 by *par-2(RNAi)* yielded α ~2, consistent with the existence of positive cooperativity (Fig. 1G). We obtained similar data regardless of whether we performed measurements at the posterior pole, where aPAR levels are low, or in a *par-3(it71)* mutant, in which PKC-3 is absent from the plasma membrane and thus PAR-2 intrinsic behavior is isolated from aPAR feedback resulting in uniform PAR-2 (Tabuse et al, 1998) (Figs. 1G and EV1C–F; Table EV1). Finally, consistent with a role for the RING domain in driving cooperativity, introduction of the RING-disrupting mutation C56S reduced apparent cooperativity to α < 1.5 (Fig. 1G).

We conclude that the RING domain drives effective cooperativity in PAR-2 membrane binding and that this cooperativity amplifies the ability of PAR-2 to be concentrated on the posterior membrane (Fig. 1H).

## RING domain dimerization is required for positive feedback

One mechanism to generate cooperativity would be for the PAR-2 RING domain to promote its own loading onto the plasma membrane. We therefore tested whether endogenous PAR-2 could recruit an isolated RING domain into the posterior PAR domain. We first expressed a soluble form of the RING domain fused to mNeonGreen (mNG). This mNG::RING fusion not only failed to localize to the posterior PAR domain, but exhibited no detectable membrane binding, appearing identical to mNG alone (Fig. 2A). This result was consistent with prior work showing that an N-terminal fragment containing the RING domain, but lacking predicted PRBH domains 2 and 3, failed to localize to the plasma membrane (Hao et al, 2006). However, when we targeted mNG::RING to the plasma membrane via fusion to PH$_{PLC\delta1}$, it was efficiently recruited into the posterior PAR domain in a manner that depended on both an intact RING domain and the presence of endogenous PAR-2 (Fig. 2B,C). Thus, while the RING domain cannot be directly recruited from the cytoplasm to the posterior membrane by PAR-2, if it is targeted to the correct membrane compartment, it is sufficient to enable co-segregation with full-length PAR-2.

How could the RING domain facilitate its co-segregation with PAR-2 into the posterior? The PAR-2 RING domain sequence harbors a C3HC4 pattern of zinc-coordinating residues that is characteristic of RING-family E3 ligases. Structural homology modeling of the PAR-2 RING domain suggested a similarity to dimeric E3 ligases (https://swissmodel.expasy.org/) and an Alpha-Fold structure prediction for a PAR-2 RING dimer was similar to dimeric E3 RING domains, which are characterized by a four-helix bundle consisting of an N and C helix from each of the two monomers (Fig. 2D) (Jumper et al, 2021). The PAR-2 RING exhibits the expected "knobs-into-holes" pattern of conserved hydrophobic residues (L50, L54, L109, M112, L116) within the predicted four-helix bundle, mutation of which has been shown to disrupt dimerization of other RING domains (Appendix Fig. S1) (Brzovic et al, 2001; Crick, 1953; Fiorentini et al, 2020) and which are broadly conserved within the *Caenorhabditis* genus (Fig. 2E). Given previous reports of PAR-2 oligomerization (Arata et al, 2016; Motegi et al, 2011), we wondered whether dimerization of the PAR-2 RING domain could underlie the cooperative membrane binding that we observe.

To test whether the PAR-2 RING domain was sufficient for oligomerization, we purified the recombinant PAR-2 RING domain and subjected the purified RING domain to SEC-MALS to reliably determine its molecular weight and thus its oligomeric state. These data revealed concentration-dependent dimerization with the dimer fraction shifting from ~25 to ~75% over the concentration range tested (0.5–10.0 mg/ml, Fig. 2F, H). To selectively disrupt dimerization, we mutated L109, the sidechain of which lies at the heart of the hydrophobic core of the putative dimer interface, making symmetric contact with L109 from the second protomer (Fig. 2D). While technical limitations prevented analysis across the same range of concentrations as wild type, at 0.75 mg/ml the L109R mutation reduced the dimer fraction from ~35 to ~5%, which is consistent with a approximately tenfold reduction in dimer affinity (Fig. 2G,H).

Having established that L109R disrupts dimerization in vitro, we examined the effects of L109R in vivo. Quantification of membrane binding revealed that L109R reduced the M:C ratio nearly as much as C56S. L109R also showed similar M:C ratios between the polarized and uniform states and substantially reduced nonlinearity in the relationship between membrane and cytoplasmic concentrations, suggesting that disruption of the dimer interface weakens positive feedback (Fig. 2I,J; Table EV1; Appendix Fig. S1). We also tested the effects of an additional predicted interface mutation (L50R). L50R also reduced membrane binding, though to a lesser extent than L109R, and did not show any additive effects with L109R (Appendix Fig. S1).

Phenotypically, L109R mutants did not exhibit any developmental defects under otherwise wild-type conditions, and thus did not fully phenocopy C56S, which showed significant levels of maternal-effect embryonic lethality and sterility, consistent with improper germline specification (Fig. 2L). However, both alleles exhibited similar maternal-effect embryonic lethality in a *nop-1(RNAi)* background in which symmetry-breaking was rendered dependent on the PAR-2 pathway due to a reduction in cortical flows. In our conditions, 100% of *nop-1(RNAi)* embryos exhibited normal development and gave rise to fertile adults, consistent with the semi-redundant contributions of cortical flow and the PAR-2 pathway to polarization (Rose et al, 1995; Tse et al, 2012; Zonies

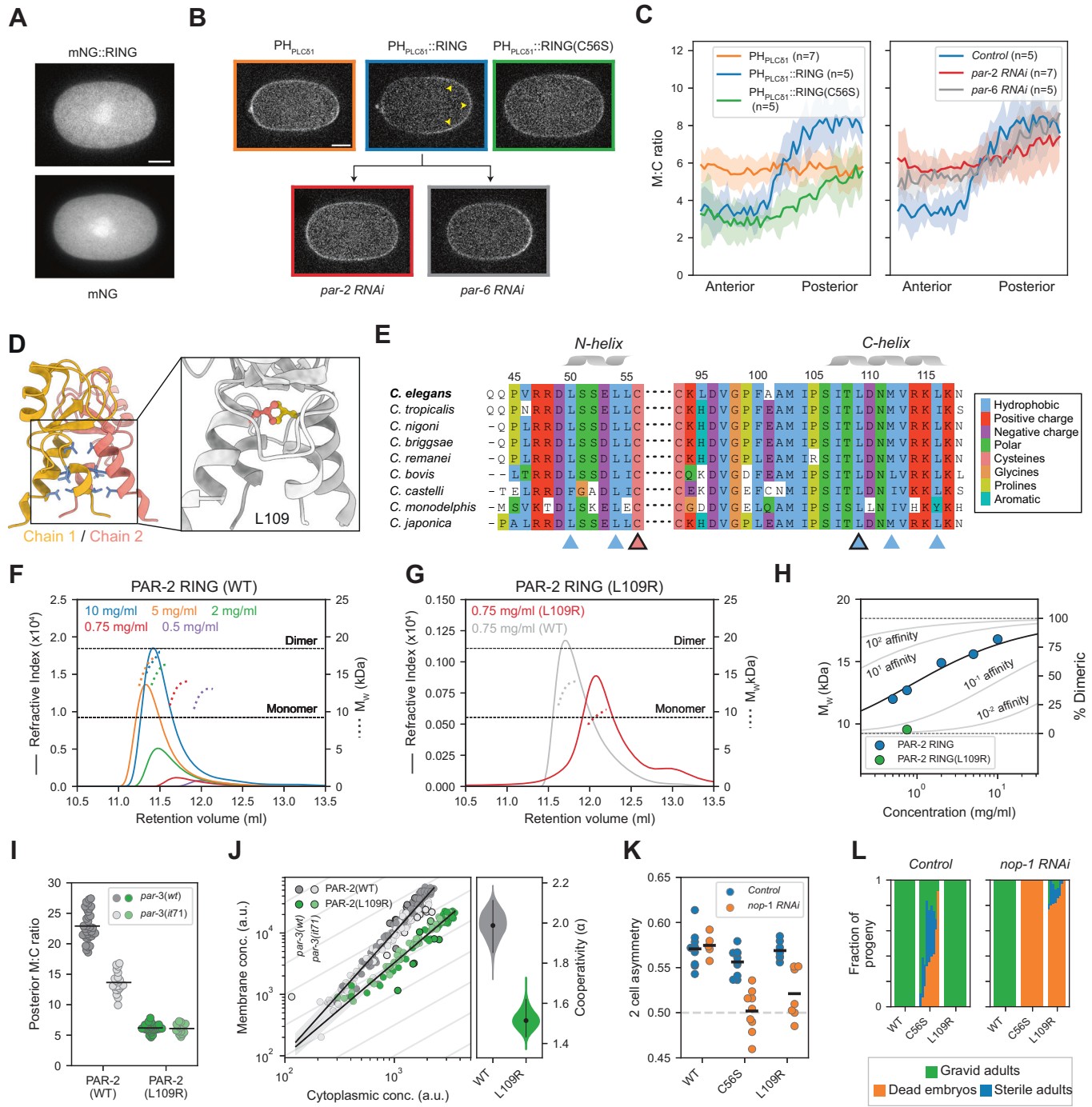

et al, 2010). By contrast, and consistent with RING mutants exhibiting defects in the PAR-2 pathway, combining *nop-1(RNAi)* with *par-2(L109R)* or *par-2(C56S)* resulted in a reduction in division asymmetry (Fig. 2K) and >80% and 100% embryonic lethality, respectively (Fig. 2L). These phenotypes are unlikely to be due to differences in protein dosage as both RING mutants were expressed ~80% of wild-type levels (Table EV1) and we did not observe synthetic lethality when we performed *nop-1(RNAi)* on heterozygous *par-2(ok1723/+)* animals, which express PAR-2 at ~60% of WT levels (Appendix Fig. S1F) (Rodrigues et al, 2023).

We conclude that dimerization of the PAR-2 RING domain underlies concentration-dependent membrane binding and is required for polarization when cortical flows are compromised and embryos rely on the PAR-2 pathway to polarize.

We do not fully understand the differences in lethality and sterility between C56S and L109R. It is likely due to the larger-scale destabilization of PAR-2 by C56S. C56S disrupts $Zn^{2+}$ chelation that is critical for RING domain folding while L109R is a surface residue at the dimer interface. Such an interpretation would be consistent with the reduced membrane affinity, slightly lower

◀

**Figure 2. Cooperative membrane association arises from dimerization of the PAR-2 RING domain.**

(A) An isolated RING domain fragment displays no membrane association. SAIBR-corrected images of an mNG-tagged PAR-2 RING domain fragment, compared to mNG alone. (B) A membrane-tethered RING domain fragment displays posterior enrichment in a polarity-dependent manner. SAIBR-corrected images of GFP::PH::RING compared to GFP::PH and GFP::PH::RING(C56S). Unlike GFP::PH and GFP::PH::RING(C56S), GFP::PH::RING displays enrichment in the posterior (arrowheads), which is compromised upon RNAi of either *par-2* or *par-6*, consistent with segregation depending on polarization of endogenous PAR-2. (C) Anterior to posterior quantification of local membrane to the cytoplasmic ratio for the strains and conditions in (B). (D) AlphaFold structure prediction for the PAR-2 RING domain dimer (PAR-2 residues 40–120), with inward-facing hydrophobics indicated (blue). Inset shows enlarged view of the 4-helix bundle with L109 indicated. (E) Clustal Omega alignments of the PAR-2 RING domain C and N helices within the *Caenorhabditis* genus. Arrowheads indicate inward-facing hydrophobic residues within the C and N helices (blue), including *C. elegans* L109 (black border). The zinc-coordinating residue C56 is also indicated (pink). (F) SEC-MALS traces for the PAR-2 RING domain at indicated input sample concentrations. Solid lines indicate differential refractive index measurements, dotted lines indicate weight-averaged molecular weight ($M_w$). Color-coded by input sample concentration. To improve visibility, $M_w$ values restricted to central peak region (RUI >80% of max). (G) Comparison of SEC-MALS traces for WT and L109R PAR-2 RING domains for input concentrations of 0.75 mg/ml. Note WT data from the 0.75 mg/ml sample in (F) reproduced to allow direct comparison. Labels as in (F). (H) The PAR-2 RING domain displays concentration-dependent dimerization. $M_w$ vs. input concentration for the SEC-MALS assays in (F, G). Solid line indicates fit of wild-type data to a dimer model (see "Methods") and is shown alongside model predictions for indicated fold-increase/decrease in affinity relative to the best fit. (I) Quantification of PAR-2(L109R) posterior membrane-to-cytoplasmic ratio in polarized (*par-3(WT)*) and uniform (*par-3(it71)*) conditions. Wild-type data from Fig. 1 repeated in gray for reference. (J) Quantification of membrane and cytoplasmic concentrations of PAR-2(L109R) in cells with varying total amounts of PAR-2 (wild-type data from Fig. 1 repeated in gray for reference). Data from both polarized cells (dark points) and uniform cells (light points) are pooled. Right: probability distribution of the cooperativity score calculated by bootstrapping. (K) Two-cell asymmetry (Area$^{AB}$/Area$^{AB+P1}$) in wild-type, *par-2(C56S)* and *par-2(L109R)* cells (control vs *nop-1 RNAi*). (L) Fraction of gravid adults, sterile adults and dead embryos in the progeny of wild-type, *par-2(C56S)* and *par-2(L109R)* worms (control vs *nop-1 RNAi*). Data Information: In (A, B), scale bars = 10 μm. (C) Mean ± SD (number of embryos, n, indicated). (I–K) Datapoints are individual embryos. (I, K) Mean indicated. (J) The best fit to the full dataset (dot) is shown with probability distribution of cooperativity (violin plot) and 95% confidence interval (lines) calculated by bootstrapping. (L) A composite of 8–10 individual trials shown for each condition. Additional statistics are available in Table EV2. Source data are available online for this figure.

overall protein amounts of C56S vs L109R (Table EV1), and the fact that attempts to purify the PAR-2(C56S) RING domain failed to yield usable quantities of intact protein. Thus, for the remainder of this work, we will limit analysis to the L109R allele.

## A simple thermodynamic model of dimerization is sufficient to generate positive feedback

To understand how the dimerization of PAR-2 generates positive feedback, we formulated a thermodynamic model based on the dimerization of a reversibly bound membrane-associated molecule. We let molecules exist in one of four states, cytoplasmic monomer, cytoplasmic dimer, membrane monomer, and membrane dimer, the relative concentrations of which will depend on the strengths of dimerization and membrane binding. Note that this model relies only on the assumptions that the molecule undergoes reversible dimerization, that dimers and monomers can reversibly associate with the membrane, and that these activities occur independently (Fig. 3A; Appendix. Supplemental Model Description).

Varying membrane ($K_D^{mem}$) and dimerization ($K_D^{dim}$) affinities revealed that cooperativity increases monotonically with membrane binding affinity (reduced $K_D^{mem}$). However, for a given $K_D^{mem}$, cooperativity peaks at an optimum value of $K_D^{dim}$, with higher or lower values reducing cooperativity (Fig. 3B,C). The region of parameter space exhibiting high cooperativity corresponded to a regime in which the dimer fraction was high at the membrane, but effectively absent in the cytoplasm (Fig. 3D,E, orange). Under such conditions, increases in local membrane concentration will tend to stabilize membrane binding via promoting dimerization.

Fits of this model to experimental RNAi rundown data for PAR-2(WT) showed good concordance. Specifically, we find that the estimated dimer affinity from model fitting ($K_D^{dim}$ ~425 nM, 95% CI [280, 654], Fig. 3F; Appendix Fig. S2) reproduces the affinity that was independently measured in vitro by analytical ultracentrifugation ($K_D^{dim}$ (global fit) = 358 nM, Fig. EV2). This was substantially higher than estimated cytoplasmic PAR-2 concentrations (10–50 nM) (Goehring et al, 2011b; Gross et al, 2019),

consistent with PAR-2 being primarily monomeric in the cytoplasm and with our observation that only a membrane-tethered form of the isolated RING domain was able to co-segregate with endogenous PAR-2 into the posterior PAR domain (Fig. 2A–C). Simultaneous fitting of both PAR-2(WT) and PAR-2(L109R) with a common $K_D^{mem}$ indicate that L109R results in an approximately sixfold reduction in dimerization affinity, which is generally consistent with the reduction in magnitude observed by SEC-MALS (Fig. 3F; Appendix Fig. S2). Constraining fits with measured values of $K_D^{dim}$ for PAR-2(WT) yielded similar results (Appendix Fig. S2).

Thus, a simple thermodynamic model of dimerization and membrane binding is sufficient to capture the cooperative membrane binding of PAR-2.

## Constitutive dimerization disrupts plasma membrane selectivity and PAR-2 function

A key prediction of our model is that both increasing or decreasing dimer affinity should compromise PAR-2 function. We have already shown that reduced dimer affinity compromised PAR-2 membrane binding and the robustness of polarization. To examine the effects of enhanced dimer affinity, we created a constitutive PAR-2 dimer by introducing a dimeric GCN4 leucine zipper at the end of the RING domain (40–120), before the first PRBH domains (Harbury et al, 1993; Illukkumbura et al, 2023). Similarly to PAR-2(WT), we found that PAR-2(GCN4) was segregated into the embryo posterior, but showed residual membrane localization in the anterior membrane, suggesting it was less sensitive to removal by aPKC (Fig. 4A,B). Unexpectedly, it also exhibited prominent accumulation on internal vesicular-like structures and a corresponding reduction in plasma membrane concentrations, suggesting that the normal preferential localization of PAR-2 to the plasma membrane is disrupted by constitutive dimerization (Fig. 4Aii,iv). These accumulations did not reflect internalized plasma membrane as they were not labeled by the plasma membrane marker, PH$_{PLC\delta1}$ (Fig. EV3A) (Kachur et al, 2008). Instead, the enrichment of PAR-2(GCN4) near the centrosomes resembled known localization of

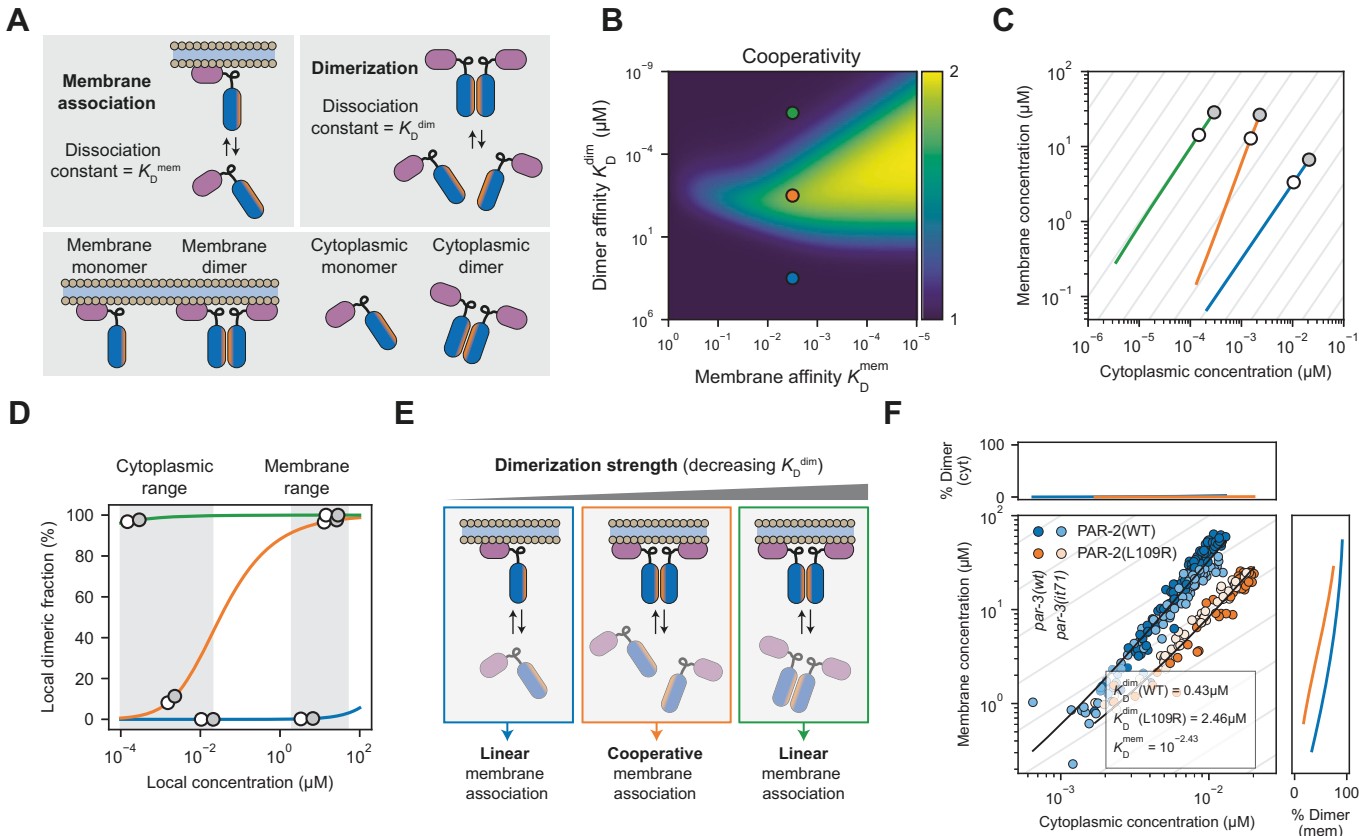

**Figure 3. Cooperativity arises from the selective stabilization of dimers at the plasma membrane.**

(A) Schematic for model of dimerization and membrane association. The model has two dissociation constants $K_D^{mem}$ and $K_D^{dim}$ representing the strength of membrane association and dimerization, respectively. (B) Membrane binding cooperativity as a function of dissociation constants $K_D^{mem}$ and $K_D^{dim}$, showing that cooperativity is optimized at intermediate dimerization strengths. Points in blue, orange and green correspond to parameter regimes shown in (C). (C) Mapping between cytoplasmic and membrane concentrations in systems with varying levels of total protein for the three parameter regimes indicated in (B). Gray and white points indicate systems with 100% and 50% total protein, respectively. (D) Degree of dimerization as a function of local concentration for protein with high (green), intermediate (orange) and low (blue) dimerization strengths (corresponding to the $K_D^{dim}$ values shown in B). Local cytoplasmic and membrane concentrations corresponding to the gray and white points in (C) are shown for reference. (E) Cooperativity arises from selective stabilization of dimers at the membrane. Where dimerization strength is intermediate, proteins exist largely as monomers in the weakly concentrated cytoplasm, but dimerize upon membrane association due to an increase in local concentration, which further stabilizes membrane association. Systems in which dimer affinity is too low or too high are not induced to change dimeric state upon membrane binding, so membrane association is linear. (F) Fit of measured membrane vs cytoplasm relationships for PAR-2(WT) and PAR-2(L109R) to a model with shared $K_D^{mem}$ and different $K_D^{dim}$ for WT vs L109R. Panels top and right show degree of dimerization in the model as a function of local concentration across the relevant range of cytoplasmic and membrane concentrations. Data Information: In (F), datapoints are individual embryos. Black lines are fits of data to a nonlinear regression model. Fraction dimer in the cytoplasm and membrane are calculated from the model. Source data are available online for this figure.

endosomal compartments (Hyenne et al, 2012; Zhang et al, 2008). Consistent with this, PAR-2(GCN4) exhibited partial colocalization with RAB-7 and to a lesser extent RAB-5 (Fig. EV3B; Appendix Figs. S3 and S4). This effect was not due to aberrant membrane targeting by the GCN4 sequence as an mNG::GCN4 fusion was diffusely localized in the cytoplasm (Fig. 4C). Finally, to validate that this effect was not specific to GCN4, we introduced an alternative dimerization motif (6HNL, (Chang and Dickinson, 2022)), which yielded similar increases in residual anterior membrane localization and accumulation on internal membranes (Fig. EV3C,D).

While *par-2(GCN4)* animals did not show significant lethality under normal conditions, when we blocked cortical flows via depletion of a myosin regulatory light chain using *mlc-4(RNAi)*, the efficiency of polarization was reduced, consistent with defects in the PAR-2 pathway (Fig. 4D,E). While embryos were often capable of generating some

asymmetry, PAR-2 domains were substantially less pronounced and were accompanied by significant levels of residual membrane-associated PAR-2 in the anterior. Thus, somewhat counter-intuitively, increasing dimerization strength reduced the ability of PAR-2 to be targeted to the posterior plasma membrane during polarization. Because both increasing and decreasing dimer affinity disrupted polarization under conditions in which the PAR-2 pathway is required, we conclude that PAR-2-dependent polarization relies on optimization of RING dimer affinity.

## Enhanced membrane affinity of ectopic PAR-2 dimers leads to kinetic trapping on inappropriate membranes

How can we explain the loss of plasma membrane specificity of PAR-2(GCN4)? Plasma membrane selectivity for proteins like PAR-2 that bind nonspecifically to negatively charged lipids is

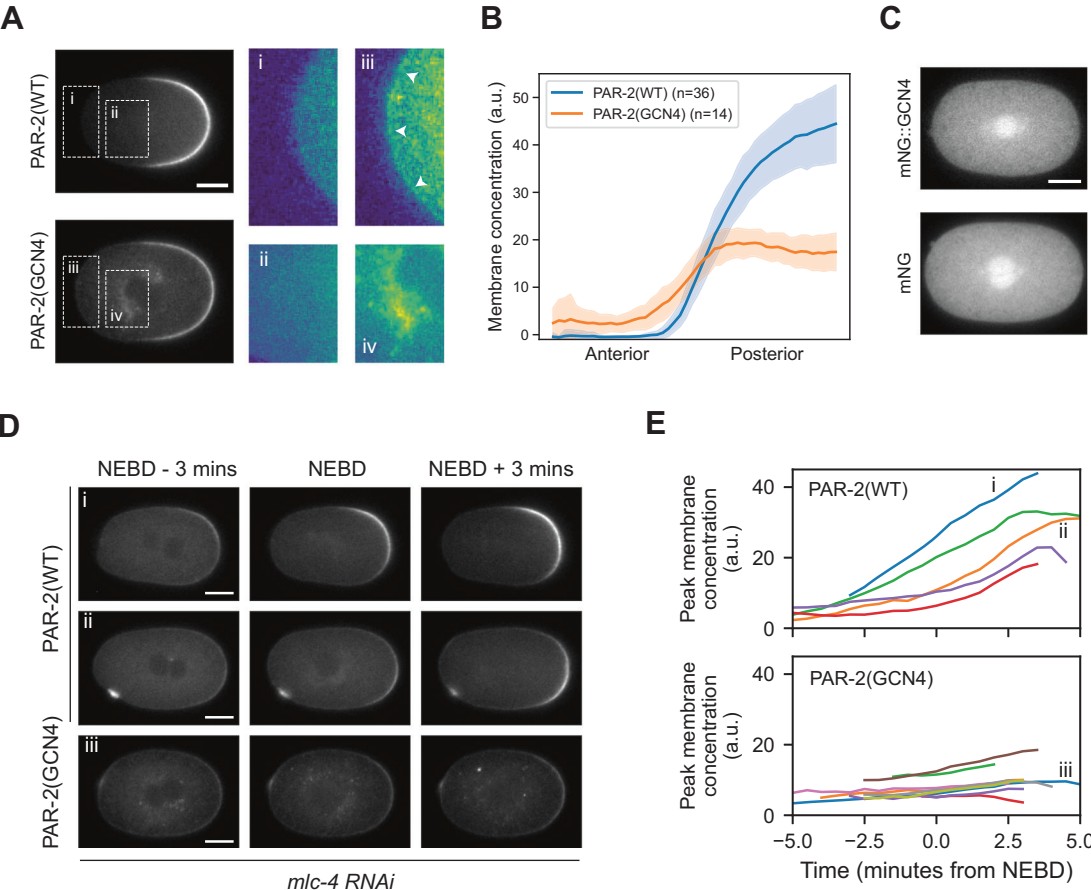

**Figure 4.  Enhanced dimerization leads to loss of membrane specificity and reduced sensitivity to PKC-3.**

(A) SAIBR-corrected images of PAR-2 and PAR-2(GCN4). Note the localization of PAR-2(GCN4) to the anterior membrane (iii, arrowheads), as well as prominent localization to internal structures (iv). (B) Anterior to posterior membrane concentration profiles of PAR-2(WT) and PAR-2(GCN4). (C) GCN4 alone does not localize to the plasma membrane or internal structures. SAIBR-corrected image of mNG::GCN4 compared to mNG alone. (D) PAR-2(GCN4) displays defective symmetry-breaking in *mlc-4 RNAi* conditions. SAIBR-corrected images of PAR-2(WT) and PAR-2(GCN4) at three minutes pre-NEBD (nuclear envelope breakdown), NEBD and three minutes post-NEBD in *mlc-4 RNAi* conditions. Wild-type PAR-2 shows variation in the timing and degree of symmetry-breaking (i vs ii). (E) Quantification of peak membrane concentration over time for PAR-2(WT) and PAR-2(GCN4) in *mlc-4 RNAi* conditions. i, ii, and iii correspond to the embryos in (D). Data Information: Scale bars in (A, C, D) = 10 µM. (B) Mean ± SD (number of embryos, *n*, indicated). Source data are available online for this figure.

generally thought to rely on the differential (i.e., more negative) charge profile of the plasma membrane relative to internal membranes rather than recognition of specific lipid species (Yeung et al, 2008). We therefore hypothesized that constitutive dimerization may provide a sufficient avidity enhancement to enable binding of PAR-2(GCN4) to endolysosomal membrane compartments despite a weaker negative charge on these membranes. Consistent with generally tighter membrane binding, PAR-2(GCN4) exhibits substantially reduced turnover at the plasma membrane (Illukkumbura et al, 2023).

We therefore introduced a second internal membrane compartment to the model with a reduced binding affinity ($K_\mathrm{D}^\mathrm{int}$), reflecting the normal preference of PAR-2 for the plasma membrane. We found that increasing dimerization generally favors partitioning to membrane compartments, leading to increased concentrations on internal membranes, consistent with dimer-dependent stabilization (Fig. 5A). However, we did not observe an enhancement of partitioning to internal membranes at the expense of plasma

membrane targeting as we observed for PAR-2(GCN4) in vivo. Instead, the relative preference for the plasma membrane increased with dimerization affinity. Thus, from an equilibrium perspective, an increase in dimerization affinity cannot explain the observed decrease in plasma membrane selectivity.

One aspect of our system we have so far largely ignored is the relevant timescale of polarization, which is neglected in analysis of equilibrium conditions. During polarization, PAR-2 must shift from being nearly fully excluded from the plasma membrane by the activity of PKC-3 at the end of meiosis II to being enriched on the posterior membrane as PKC-3 is segregated into the anterior at the start of mitosis—a span of ~10 min (Fig. 5B) (Cuenca et al, 2003; Reich et al, 2019). These dynamics place temporal constraints on membrane binding—if membrane affinity is too high, redistribution of PAR-2 between membrane compartments may simply be too slow, leaving PAR-2 kinetically trapped on internal membranes. Consistent with this picture, we found that PAR-2(GCN4) exhibits both reduced mobility in the cell interior and slower redistribution

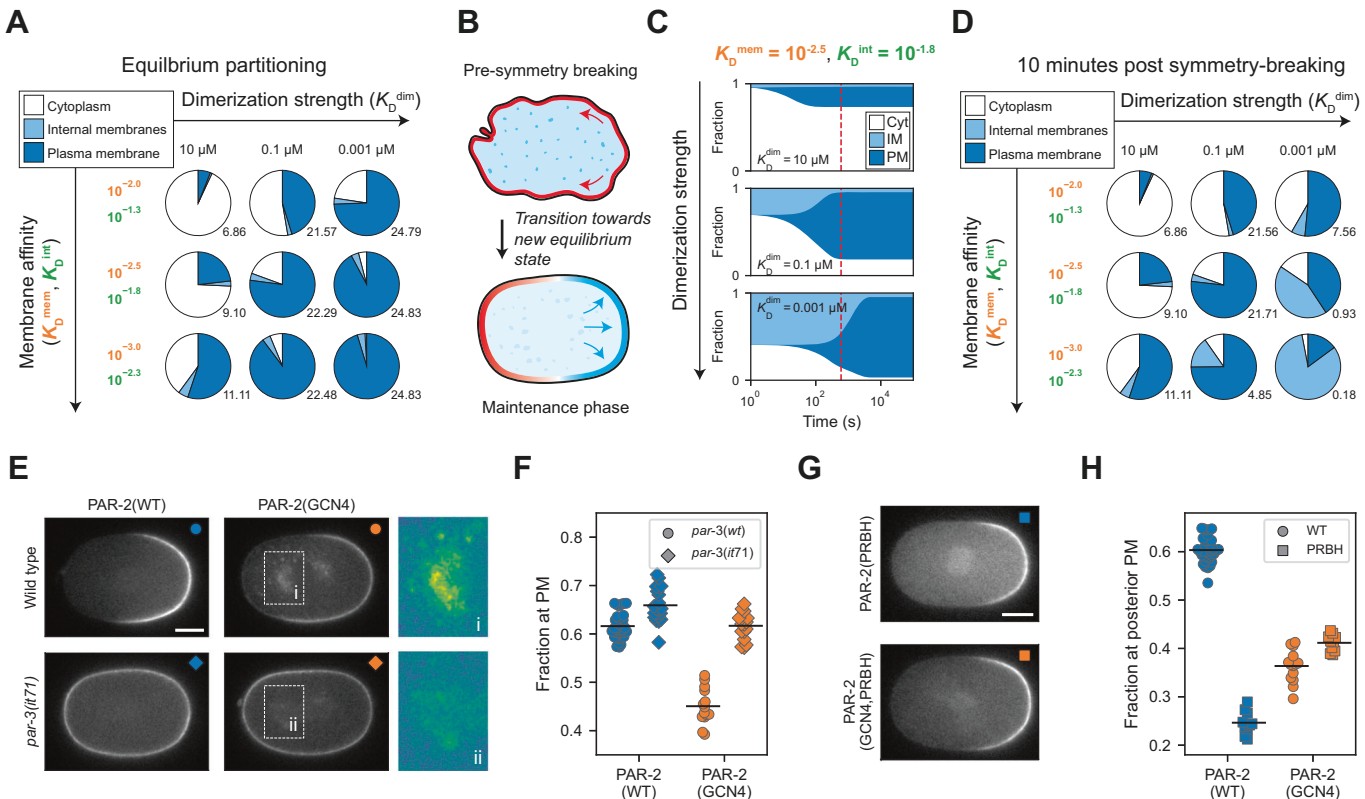

**Figure 5. Intermediate dimerization affinity optimizes exchange kinetics to ensure membrane specificity.**

(A) Equilibrium solutions for a three-compartment model (cytoplasm, internal membranes (IM), and plasma membrane (PM) as a function of dimerization strength ($K_D^{dim}$) and membrane affinity ($K_D^{mem}$ and $K_D^{int}$). Pie charts show the fraction of total protein in each compartment at equilibrium. Numbers to the bottom right of each chart show the ratio of protein in the PM and IM compartments. $K_D^{int}/K_D^{mem}$ is fixed to a constant ratio of 5. (B) Schematic for symmetry-breaking described as a transition between two equilibrium states. The system begins in a pre-symmetry-breaking state in which only the cytoplasm and internal membrane compartments are available to PAR-2, with PAR-2 being held off the PM by the uniform activity of aPARs (red). At symmetry-breaking, transport of aPARs to the anterior relieves inhibition of PAR-2 at the posterior plasma membrane, and the system transitions towards a new state reflecting this change in membrane availability. (C) Fraction of protein in each of the three compartments over time as the system transitions between a two compartment model (cytoplasm and IM only) and a three-compartment model (PM added), as a function of dimerization strength. When dimerization is weak (top), the system rapidly reaches a new equilibrium, but with weak PM association. When dimerization is strong (bottom), the final equilibrium state has strong PM association, but the system takes exponentially longer to reach this state. With intermediate dimerization (middle), an intermediate behavior is observed. Dashed red line marks 600 s, roughly corresponding to the timing of NEBD after symmetry-breaking. (D) PM specificity 10 min post symmetry-breaking is maximized in intermediate dimerization regimes. As in (A), but showing a snapshot 10-min into the transition between the pre-symmetry-breaking and maintenance phase states. Middle row corresponds to the simulations in (C). Note that in high dimerization regimes, PM association and specificity is maximized when membrane affinity is reduced as a result of increased transition kinetics. (E) SAIBR-corrected images of PAR-2 and PAR-2(GCN4) in *par-3(WT)* and *par-3(it71)* backgrounds. Note the reduced IM association of PAR-2(GCN4) in *par-3(it71)* conditions (ii vs i). (F) Fraction of total PAR-2 at the PM in each of the four conditions in (E). Note in contrast to (H), here we use fraction at PM, rather than fraction at posterior PM, as we are comparing polarized and unpolarized conditions. (G) SAIBR-corrected images of PAR-2(PRBH) and PAR-2(GCN4,PRBH). Note the lack of IM association in PAR-2(GCN4,PRBH) (compare to (E), top row). (H) Fraction of total protein at the posterior PM for PAR-2(WT), PAR-2(GCN4), PAR-2(PRBH) and PAR-2(GCN4,PRBH). Whereas mutating PRBH residues in PAR-2(WT) strongly decreases posterior plasma membrane association, adding these mutations to PAR-2(GCN4) leads to a moderate *increase* in posterior plasma membrane association. Data Information: Scale bars in (E, G) = 10 µM. (F, H) Datapoints represent individual embryos. All data shown, mean indicated. Additional statistics, including analysis of effect size, are available in Table EV2. Source data are available online for this figure.

from the cell interior onto the posterior plasma membrane during polarization (Fig. EV4).

To explore how dimerization affinity influenced polarization timescales in our model, we used transition state theory to assess the time evolution of our dual membrane system. Specifically, we examined the shift from an unpolarized equilibrium state in which molecules only have access to the low-affinity internal membrane compartment, and a polarized equilibrium state in which molecules gain access to the plasma membrane, reflecting the clearance of PKC-

3 from the posterior membrane of the zygote during polarization. For all values of $K_D^{dim}$, molecules were initially excluded from the plasma membrane and then relocalized to the plasma membrane over time at the expense of both the cytoplasmic and internal membrane compartments. Importantly, all eventually reached a steady state in which concentrations at the plasma membrane exceeded that on internal membranes, reflecting the differential affinities for the two membrane compartments (Fig. 5C). However, as we suspected, given the stabilizing effect of dimerization on the membrane binding,

increasing dimerization strength dramatically slowed the timescale of this redistribution from internal pools to the plasma membrane in the model (Fig. 5C; Appendix Fig. S5). Consequently, if assessed at intermediate timepoints (e.g., ~10 min), increasing dimerization affinity appears to enhance the internal membrane pool at the expense of the plasma membrane (Fig. 5D). Note, this apparent loss of selectivity arises purely from slower kinetics caused by dimerization-dependent reduction of membrane dissociation such that at similar timepoints, the strong dimer system is much further from the equilibrium, plasma membrane dominated state.

This model therefore predicts that kinetic trapping of PAR-2 on internal membranes will be reduced if we either extend the time available for PAR-2 to equilibrate between the internal and plasma membrane compartments or compensate for the increase in avidity provided by enhanced dimerization by reducing the affinity of monomers for membranes. To increase the time available for equilibration, we examined the behavior of PAR-2(GCN4) in *par-3(it71)* embryos, which lack PKC-3 activity at the membrane and thus PAR-2 is not cleared from the membrane at the end of meiosis II (Reich et al, 2019; Tabuse et al, 1998). Consistent with predictions, PAR-2(GCN4) exhibited reduced levels of localization to internal membranes in *par-3(it71)* embryos compared to *par-3(wt)* embryos (Fig. 5E,F).

To ask whether reduction of membrane affinity of monomers could rescue kinetic trapping, we targeted one of three putative PRBH motifs that are thought to mediate PAR-2 membrane binding (Bailey and Prehoda, 2015; Brzeska et al, 2010; Ramanujam et al, 2018) (Fig. 1B). As replacement of seven serines to glutamic acid is sufficient to prevent PAR-2 enrichment at the plasma membrane (Hao et al, 2006), we introduced two S>E mutations into the PRBH3 region (S334E, S338E) of wild-type and constitutively dimeric PAR-2, yielding PAR-2(PRBH) and PAR-2(PRBH, GCN4), respectively. Consistent with reduced kinetic trapping, we found that PAR-2(PRBH, GCN4) exhibited reduced accumulation on internal membranes compared to PAR-2(GCN4) (Fig. 5G vs E) and restored near wild-type rates of cytoplasmic turnover (Fig. EV4). PAR-2(PRBH, GCN4) also exhibited none of the residual anterior localization seen for PAR-2(GCN4), suggesting reduced affinity also restored normal sensitivity to PKC-3 (Figs. 5G vs E and 4A). Finally, whereas PAR-2(PRBH) exhibited reduced accumulation within the posterior PAR domain relative to PAR-2(WT), consistent with what one would expect for reduced membrane affinity (Fig. 5G,H, PRBH), not only does PAR-2(PRBH, GCN4) not exhibit reduced accumulation relative to PAR-2(GCN4), but actually accumulated to slightly higher levels (Fig. 5H). This somewhat counterintuitive effect of affinity-reducing mutations in the context of the constitutive dimer is consistent with reduced plasma membrane affinity of PAR-2(PRBH, GCN4) being balanced out by reduced kinetic trapping and enhanced redistribution from internal pools to the plasma membrane. Taken together, these results indicate that one can at least partially rescue the effects of constitutive dimerization (decreased $K_D^{dim}$) by reducing the intrinsic membrane binding affinity of the constituent monomers (increased $K_D^{mem}$) (Fig. 5D).

We therefore conclude that constitutive dimerization kinetically traps PAR-2 on internal membranes through enhanced membrane binding, substantially increasing the timescale required for plasma membrane accumulation during polarization.

The fact that the robustness of polarization by the PAR-2 pathway is compromised by both increases and decreases in dimer affinity strongly suggests that dimerization affinity has been optimized. Such optimization ensures that membrane binding of PAR-2 is both sufficiently cooperative to support robust polarization, but also sufficiently dynamic that PAR-2 remains highly responsive to spatiotemporal changes to the system, such as those involved in the meiosis-mitosis transition and polarization.

## Discussion

Here we have identified cooperative membrane binding of PAR-2 as a key feature of the PAR polarity network in *C. elegans* and directly linked this behavior to optimized dimerization of its RING domain.

Although the vast majority of RING domain-containing proteins identified in humans are believed to act as E3 ubiquitin ligases (Deshaies and Joazeiro, 2009), a key role of RING domains is to facilitate protein-protein interactions (Borden, 2000). In the case of E3 ligases, RING domains typically recruit E2 ubiquitin conjugating enzymes to facilitate substrate modification. Importantly for our work, E3 ligases often act as multimers in which RING-RING interactions play critical roles in mediating interactions between E3 monomers or in E2 recruitment (Fiorentini et al, 2020). While we cannot rule out that PAR-2 may possess E3 ubiquitin ligase activity, our data suggest that it is this dimerization function of the RING domain that is critical in defining the cooperative nature of PAR-2 membrane binding.

Specifically, we show that membrane binding cooperativity emerges from the optimization of dimer affinity such that the $K_D$ is intermediate between the effective cytoplasmic and membrane concentrations. Consistent with this model, both increasing or decreasing dimerization affinity impacted the ability of PAR-2 to polarize. RING domains of E3 ligases can exhibit a broad range of dimer affinities (Fiorentini et al, 2020) and, analogously to what we have shown here, differences in RING dimer affinity in E3 ligases have been proposed to underlie distinct modes of substrate binding and activity regulation (Koliopoulos et al, 2016). We therefore suggest that the RING domain provides a highly tunable platform for dimer optimization, a feature which in this case appears to have been co-opted to facilitate robust polarization of the PAR polarity network.

Cooperativity does not arise from direct recruitment of cytoplasmic monomers by membrane-associated species, which is negligible in this system due to the low concentration of cytoplasmic molecules, a conclusion supported by the failure of soluble isolated RING domains to be recruited by PAR-2 to the posterior domain. Rather, effective positive feedback arises because local increases in membrane concentration will favor dimerization of membrane-associated monomers, which will in turn render them more stably associated with the membrane (Agudo-Canalejo et al, 2020). It is therefore specifically membrane-dependent dimerization that accounts for the observed positive feedback.

Reinforcing the need to optimize dimer affinity, increasing dimer affinity led to loss of plasma membrane selectivity. We initially considered that the impact of dimerization on nonlinear dynamics might lead to a reduction in the relative preference of

PAR-2 for the plasma vs. internal membranes. However, increasing dimerization favored plasma membrane binding in our equilibrium model. Instead, we found that loss of selectivity was due to a kinetic mismatch between the timescales of polarization and membrane turnover of the constitutive dimer on membranes. Due to enhanced stabilization of membrane binding by constitutive dimers, redistribution of PAR-2(GCN4) dimers from internal membrane pools to the plasma membrane at the onset of polarization is simply too slow. Even in the absence of large scale reorganization, such dynamic redistribution is likely to be required to counter internalization of membrane-associated molecules by endocytosis and may explain why we observe some level of internal membrane binding even when PAR-2(GCN4) is rendered resistant to membrane displacement by aPARs (e.g., in *par-3* embryos).

Our in vitro analysis of the isolated RING domain revealed no signs of higher-order oligomer formation beyond dimers. This contrasts with prior work based on single particle tracking of PAR-2 particles that suggested a roughly even mix of oligomer sizes from 1 to 4 (Arata et al, 2016). It is possible that other regions of PAR-2 could mediate higher-order assemblies, which could explain residual cooperativity observed in RING mutants. That said, the agreement between our in vitro and in vivo measurements of RING dimer affinity suggests that one need not invoke the existence of larger oligomers to explain the observed cooperativity. As previous work did not address the mechanism or kinetics of oligomerization or confirm particle size estimates obtained from single particle tracking via other methods, further work will be required to clarify the existence and potential roles for PAR-2 oligomers of size >2.

We would also stress that the cooperativity we observe cannot, on its own, account for polarization of the embryo. Indeed, the thermodynamic equilibrium model we used to fit our experimental data cannot break the symmetry or sustain a polarized steady state by construction due to lack of energy input. To fully capture the process of polarization by the PAR-2 pathway, cooperative membrane binding of PAR-2 must be embedded within a larger network that takes into account active ATP-dependent processes, including phosphorylation cycles driven by PKC-3. This makes sense as PAR-2 domain formation absolutely requires its phosphorylation by PKC-3 in the zygote and other P lineage cells (Hao et al, 2006; Hubatsch et al, 2019; Motegi et al, 2011; Ng et al, 2023). Ultimately, introducing cooperativity into models including energy-dependent feedback circuits will likely enhance the pattern-forming capabilities of such systems (Diambra et al, 2015; Lang and Munro, 2022). Under normal conditions, this cooperative membrane binding of PAR-2 binding plays a supporting role in stabilizing the initial aPAR asymmetry induced by cortical actomyosin flows, likely explaining the viability of *par-2(L109R)* mutants. However, as we show, this cooperativity becomes essential when cortical flows fail and polarity is initiated by PAR-2 domain formation.

It has been speculated that membrane-stabilized oligomeric assemblies can constitute an effective memory in polarizing systems (Illukkumbura et al, 2023; Lang and Munro, 2022). Specifically, by slowing the timescale of membrane turnover, oligomerization can amplify and lock in the effects of transient polarizing cues. However, our data suggest that this "memory" comes at the cost of reduced responsiveness of the system as stable dimers are slow to adapt to changes in cell state, such as the meiosis-mitosis transition we describe here. Our work therefore highlights how optimization of oligomerization kinetics, in this case of a dimeric RING domain, allows systems to balance this trade-off between memory and responsiveness in a dynamic system, which in the case of the PAR network facilitates robust and timely polarity establishment. Notably, both reduction and enhancement of dimer affinity impair the ability of PAR-2 to respond to symmetry-breaking cues, resulting in defects in the PAR-2-dependent polarization pathway. Given the widespread occurrence of dynamic oligomerization and oligomerization-dependent localization and activation within molecular networks (Hansen et al, 2022; Liau et al, 2020; Nam et al, 2007), including the PAR and other polarity networks (Benton and Johnston, 2003; Chang and Dickinson, 2022; Dodgson et al, 2013; Harris, 2017; Illukkumbura et al, 2023; Lang and Munro, 2022; Meca et al, 2019; Mizuno et al, 2003; Sailer et al, 2015; Strutt et al, 2011), this paradigm of optimized and reversible oligomerization kinetics is likely to be a broadly applicable strategy for rapid and cooperative intracellular targeting.

## Methods

**Reagents and tools table**

| Reagent/resource | Reference or source | Identifier or catalog number |
| --- | --- | --- |
| **Experimental models** | | |
| OP50: *E. coli, B, ura-* | CGC | WBStrain00041969 |
| HT115(DE3): *E. coli, F-, mcrA, mcrB, IN(rrnD-rrnE)1, rnc14::Tn10(DE3 lysogen:lavUV5 promoter-T7 polymerase)* | CGC | WBStrain00041080 |
| Rosetta(DE3): *E. coli, F- ompT hsdSB(rB- mB-) gal dcm (DE3) pRARE (CamR)* | Novagen | Cat# 70954 |
| N2: *C. elegans, Wild type* | CGC | WBStrain00000001 |
| CGC32: *C. elegans, sC1(s2023) [dpy-1(s2170) umnIs21] III].* | CGC | WBStrain00004963 |
| JK2533: *C. elegans, qC1 dpy-19(e1259) glp-1(q339)[qIs26] III/eT1 (III;V)* | CGC | WBStrain00022579 |
| KK571: *C. elegans, lon-1(e185) par-3(it71)/qC1 dpy-19(e1259) glp-1(q339)III* | Cheng et al, 1995 | WBStrain00023571 |
| LP637: *C. elegans, par-2(cp329[mNG-C1^PAR-2]) III* | Dickinson et al, 2017 | WBStrain00024329 |
| NWG0170: *C. elegans, par-2(ok1723) / sC1(s2023) [dpy-1(s2170) umnIs21] III* | Rodrigues et al, 2023 | NWG0170 |
| NWG0201: *C. elegans, par-2(cp329[mNG-C1^PAR-2]); lon-1(e185) par-3(it71) / qC1 dpy-19(e1259) glp-1(q339) [qIs26] III* | This study | NWG0201 |
| NWG0240: *C. elegans, par-2(crk41[mNG::par-2(C56S)] *cp329) / sC1(s2023) [dpy-1(s2170) umnIs21] III]* | This study | NWG0240 |
| NWG0246: *C. elegans, par-2(crk41[mNG::par-2(C56S)] *cp329); lon-1(e185) par-3(it71) / qC1 dpy-19(e1259) glp-1(q339)[qIs26] III* | This study | NWG0246 |

| Reagent/resource | Reference or source | Identifier or catalog number |
|---|---|---|
| NWG0313: C. elegans, crkSi4[pTB005: mex-5p::GFP::PAR-2(1-177)::PH::nmy-2UTR] | This study | NWG0313 |
| NWG0325: C. elegans, par-2(cp329[mNG-C1^PAR-2]) / sC1(s2023) [dpy-1(s2170) umnIs21] III] | This study | NWG0325 |
| NWG0338: C. elegans, par-2(crk82[mNG::par-2(L109R)]*cp329) / sC1(s2023) [dpy-1(s2170) umnIs21] III] | This study | NWG0338 |
| NWG0347: C. elegans, glh-1(crk148[glh-1::T2A::mNG::INPP4A]) I | This study | NWG0347 |
| NWG0351: C. elegans, par-2(crk89[mNG::par-2(L50R)]*crk82) / sC1(s2023) [dpy-1(s2170) umnIs21] III] | This study | NWG0351 |
| NWG0369: C. elegans, par-2(crk82[mNG::par-2(L109R)]*cp329) lon-1(e185) par-3(it71) / qC1 dpy-19(e1259) glp-1(q339)[qIs26] III | This study | NWG0369 |
| NWG0373: C. elegans, crkSi5[pTB006: mex-5p::GFP::PAR-2(1-177, C56S)::PH::nmy-2UTR] | This study | NWG0373 |
| NWG0374: par-2(crk104[mNG::par-2(S241A)]*cp329) | Illukkumbura et al, 2023 | NWG0374 |
| NWG0376: C. elegans, par-2(crk106[mNG::par-2(GCN4_IL)]*cp329) / sC1(s2023) [dpy-1(s2170) umnIs21] III] | This study | NWG0376 |
| NWG0378: C. elegans, glh-1(crk150[glh-1::tPT2A::mNG::INPP4A]) I | This study | NWG0378 |
| NWG0383: C. elegans, glh-1(crk151[glh-1::tPT2A::mNG::GCN4(IL)]) I | This study | NWG0383 |
| NWG0400: C. elegans, par-2(crk114[mNG::par-2(L50R)]*cp329) / sC1(s2023) [dpy-1(s2170) umnIs21] III] | This study | NWG0400 |
| NWG0407: C. elegans, par-2(crk120[mNG::par-2(C56S, L109R)]*cp329) / sC1(s2023) [dpy-1(s2170) umnIs21] III] | This study | NWG0407 |
| NWG0421: C. elegans, glh-1(crk153[glh-1::tPT2A::mNG::par-2(1-177)]) I | This study | NWG0421 |
| NWG0437: C. elegans, par-2(crk130[mNG::par-2(GCN4_IL)]*cp329) lon-1(e185)par-3(it71)/qC1dpy-19(e1259)glp-1(q339)[qIs26] III | This study | NWG0437 |
| NWG0481: C. elegans, par-2(crk106[mNG::par-2(6HNL)]*cp329) | This study | NWG0481 |
| NWG0489: C. elegans, par-2(crk170[mNG::par-2(GCN4_IL, S334E, S338E)]*crk106) / sC1(s2023) [dpy-1(s2170) umnIs21] III] | This study | NWG0489 |
| NWG0495: C. elegans, par-2(crk171[mNG::par-2(S334E, S338E)]*cp329) / sC1(s2023) [dpy-1(s2170) umnIs21] III] | This study | NWG0495 |

| Reagent/resource | Reference or source | Identifier or catalog number |
|---|---|---|
| NWG0578: C. elegans, ltIs44pAA173; [pie-1p-mCherry::PH(PLC1delta1)+unc-119(+)] V.; par-2(crk106[mNG::::par-2(GCN4_IL)]*LP637) / sC1(s2023) [dpy-1(s2170) umnIs21] III] | This study | NWG0578 |
| OD58: C. elegans, unc-119(ed3) III; ltIs38[pAA1; pie-1::GFP::PH(PLC1 d1) + unc-119(+)] | Audhya et al, 2005 | WBStrain00029210 |
| OD70: C. elegans, unc-119(ed3) III; ltIs44pAA173; [pie-1p-mCherry::PH(PLC1delta1)+unc-119(+)] V.; | Kachur et al, 2008 | WBStrain00029215 |
| SV2061: C. elegans, ttTi5605(he314[Ppie-1::glo-epdz::mcherry(smu-1)::tbb-2(3'UTR)]) II; cxTi10816(he259[Peft-3::ph::co-egfp::co-lov::tbb-2(3'UTR)]) IV | Fielmich et al, 2018 | WBStrain00051046 |
| **Recombinant DNA** | | |
| RNAi Feeding clone: xfp | C. Eckmann | |
| RNAi Feeding clone: par-2 | Source BioScience | WB Clone: sjj_F58B6.3 |
| RNAi Feeding clone: par-6 | Source BioScience | WB Clone: sjj_T26E3.3 |
| RNAi Feeding clone: nop-1 | Source BioScience | WB Clone: sjj_F25B5.2 |
| RNAi Feeding clone: mlc-4 | Redemann et al, 2010 | |
| pRI021 (ttTi5605 Mos1 insertion vector (mex-5 promoter and nmy-2 3' UTR)) | This study | |
| pDD122 (Cas9 + sgRNA plasmid for ttTi5605 Mos1 insertion) | Addgene, Dickinson et al, 2013 | Cat # 47550 |
| pETM11-SUMO3eGFP (His-SUMO bacterial expression vector) | EMBL | |
| **Antibodies** | | |
| Mouse-anti-mNeonGreen antibody | ChromoTek | Cat# 32f6, RRID:AB_2827566 |
| Rabbit-anti-Rab-5 antibody | Poteryaev et al, 2007 | |
| Rabbit-anti-Rab-7 antibody | Poteryaev et al, 2007 | |
| Rabbit-anti-Rab-11 antibody | Poteryaev et al, 2007 | |
| **Oligonucleotides and sequence-based reagents** | | |
| Synthetic DNA | This study | Table EV1 |
| CRISPR Repair Templates | This study | Table EV1 |
| sgRNA | This study | Table EV1 |
| PCR Primers | This study | Table EV1 |
| **Chemicals, enzymes and other reagents** | | |
| **Software** | | |
| AlphaFold Colab (AlphaFold v2.1.0) | https://github.com/google-deepmind/alphafold/blob/91b43223422420d1783ed802c8b3a8382a9309fd/notebooks/AlphaFold.ipynb | |
| Fiji v2.14.0 | https://imagej.net/software/fiji/, Schindelin et al, 2012 | RRID:SCR_002285 |

| Reagent/resource | Reference or source | Identifier or catalog number |
|---|---|---|
| Python v3.11.6 | see https://github.com/goehringlab/2024-Bland-EMBO/blob/main/requirements.txt | RRID:SCR_008394 |
| SAIBR Python v0.1.6 | https://github.com/goehringlab/saibr_fiji_plugin, Rodrigues et al, 2022 | RRID:SCR_024804 |
| SEDPHAT | Vistica et al, 2004 | RRID:SCR_016254 |
| Other | | |

## Methods and protocols

### C. elegans—strains and culture conditions

*C. elegans* strains were maintained at 20 °C on nematode growth media (NGM) seeded with OP50 bacteria (Stiernagle, 2006). Strains listed in Reagents and Tools Table. Use of strains is detailed in Table EV3.

## Strain construction

To generate point mutations or small insertions, mutation by CRISPR/Cas9 was performed using the protocol described by Arribere et al (2014). Repair templates were designed containing the target mutation and silent restriction sites to aid screening. crRNA guides were annealed with tracrRNA (IDT) by combining 0.5 µl tracrRNA (4 mg/ml) with 2.75 µl guide (100 M) and 2.75 µl duplex buffer (IDT), and incubating at 95 °C for 5 min. Injection mixes were prepared containing the annealed crRNA/tracrRNA, repair template and Cas9 protein (IDT, 1 µL at 10 mg/ml), along with either *dpy-10* or *unc-58* co-CRISPR markers (Arribere et al, 2014). Injection mixes were incubated at 37 °C for 10 min, centrifuged at 14,000 rpm for 10 min and injected into the gonads of adult worms. Mutants were identified by PCR and verified by sequencing.

To generate the membrane-tethered RING domain construct, sequences for PH, GFP, and PAR-2(1-177) were assembled in pRI21, a vector designed for inserting genes at the ttTi5605 *mos1* locus under control of a *mex-5* promoter and an *nmy-2* 3' UTR. A C56S mutant form of the construct was generated by site-directed mutagenesis (Q5 site-directed mutagenesis kit from New England Biolabs). Insertions were performed by CRISPR using the method described by Dickinson et al (2015).

mNG, mNG::RING, and mNG::GCN4 were expressed by inserting at the 3' end of the *glh-1* gene preceded by a self-cleaving peptide using an approach similar to Goudeau et al, (2021). NeonGreen was inserted first, flanked by the self-cleaving peptide T2A and INPP4A, an optimized CRISPR guide site included to serve as a base for further insertions (Duan et al, 2020). Insertion was performed by CRISPR/Cas9 using the method described by Dokshin et al (2018). crRNA guides targeting the 3' end of *glh-1* were annealed with tracrRNA (IDT) by combining 0.5 µl tracrRNA (4 mg/ml) with 2.75 µl guide (100 µM) and 2.75 µl duplex buffer (IDT), and incubating at 95 °C for 5 min. PCR products containing the sequence to be inserted (tPT2A::mNG::INPP4A), and the same sequence with 100 bp homology arms to *glh-1* were generated and column purified (Qiagen, QIAquick PCR purification kit), mixed in equimolar amounts, denatured at 95 °C, and annealed by gradually cooling to room temperature to generate a pool of products with single-stranded DNA overhangs to act as the repair

template. The injection mix was prepared containing the annealed crRNA/tracrRNA, repair template and Cas9 protein (IDT, 1 µL at 10 mg/ml), along with a *dpy-10* co-CRISPR marker (Arribere et al, 2014). The injection mix was incubated at 37 °C for 10 min, centrifuged at 14,000 rpm for 10 min and injected into the gonads of adult worms. Mutants were identified by PCR and verified by sequencing. While this led to good expression of mNG, we found considerable expression of GLH-1::mNG fusion products resulting from incomplete ribosome skipping at T2A (Kim et al, 2011). To minimize this, we inserted an additional self-cleaving peptide, P2A, in tandem with T2A (Liu et al, 2017; Pan et al, 2017) by CRISPR/Cas9, which reduced the incidence of read-through products compared to T2A alone, without impacting expression levels. Further insertions (RING and GCN4) were performed by CRISPR/Cas9 targeted to the INPP4A site, generating N-terminal mNG fusions (mNG::RING and mNG::GCN4). Recombinant oligonucleotides are listed in Table EV4.

## RNA interference

RNAi was performed using the feeding method described in Kamath and Ahringer (2003). Bacterial feeding clones were grown in LB liquid culture with ampicillin (50 µg/ml) for 16 h at 37 °C in a shaking incubator. dsRNA expression was then induced with IPTG (5 mM), and 150 µl bacteria were struck onto 60-mm NGM agar plates, which were then incubated at room temperature for 24 h. To obtain complete depletions, L4 worms were added to plates and incubated at 20 °C for 24–48 h before imaging. To obtain graded depletions, adult/L4 worms were left on plates for 0–24 h.

## Immunofluorescence

Immunofluorescence was performed as described (Rodriguez et al, 2017) with minor modifications. Briefly, gravid hermaphrodite worms were transferred to 7 µl M9 on a 0.1% poly-lysine-coated coverslip with 15-µm polystyrene beads. Embryos were released using a needle and then covered with a slide and mild pressure applied. Slides were snap-frozen on dry ice for 30 min after which the coverslip was quickly removed and the slide fixed in pre-chilled −20 °C methanol for 15 min. Samples were washed and re-hydrated with PBS followed by two 5 min washes in PBS+0.2% Tween-20 (PBSA) and one 60-min incubation with 1% BSA in PBSA, before proceeding with antibody incubations. All antibodies used in this study are listed in the Reagents and Tools Table. Primary antibody dilutions: anti-RAB-5/7/11(rabbit) 1:400, anti-mNG(mouse) 1:400. Secondary antibody dilutions: 1:1000.

## Microscopy

Embryos were dissected in 8 µl egg buffer or Shelton's Growth Medium for meiotic embryos (Shelton and Bowerman, 1996), and mounted between a slide and coverslip with 20-µm polystyrene beads. Midplane confocal images were captured using a ×60 objective lens on a Nikon TiE system equipped with an X-Light V1 spinning disk system (CrestOptics) with 50-µm slits, Obis 488/561 nm fiber-coupled diode lasers (Coherent) and an Evolve delta camera (Photometrics). The system was controlled using Metamorph (Molecular Devices) and configured by Cairn Research. For photobleaching, embryos were mounted as above, but imaged using a ×100 1.4 NA objective. Photobleaching was performed using a 473-nm laser directed by an

iLAS targeted illumination system (Roper). A $50 \times 50$ px box was bleached in the center of the anterior cytoplasm and images captured at 0.5 s intervals. For fixed samples, a ×100 objective and ×1.5 optovar were used and $80 \times 0.25$ μm sections captured.

## Image analysis—FRAP

Fluorescence within the bleached ROI and a corresponding control ROI in the posterior cytoplasm were measured. Fluorescence in the bleached ROI was first normalized to the control ROI and then normalized to the prebleach and postbleach signals. Prebleach was defined by the mean signal of the ten frames prior to bleaching. As all embryos experienced a very rapid initial recovery phase, we defined the first postbleach frame as 1 s after bleaching to isolate long timescale kinetics.

## Image analysis—quantification of membrane and cytoplasmic concentrations

### Image preprocessing
Images were autofluorescence corrected using SAIBR (Rodrigues et al, 2022), and a 50-pixel wide line following the membrane around the embryo was computationally straightened to simplify geometry for further analysis.

### Quantification model
Our method is adapted from previous methods that model intensity profiles perpendicular to the membrane as the sum of distinct cytoplasmic and membrane signal components (Gross et al, 2019; Reich et al, 2019). Typically these two components are modeled as an error function and a Gaussian function, respectively, representing the expected shape of a step and a point convolved by a Gaussian point spread function (PSF) in one dimension. Using this model, one can generate simulated images of straightened cortices as the sum of two tensor products which represent distinct membrane and cytoplasmic signal contributions (Fig. EV5A):

$$c_{cyt} \otimes s_{cyt} + c_{mem} \otimes s_{mem}$$

where $c_{cyt}$ and $c_{mem}$ are cytoplasmic and membrane concentration profiles and $s_{mem}$ and $s_{cyt}$ are, by default, Gaussian and error function profiles. We impose the constraint that the cytoplasmic concentration $c_{cyt}$ is uniform throughout each image. Using a differentiable programming paradigm, the input parameters to the model can be iteratively adjusted by backpropagation to minimize the mean squared error between simulated images and ground truth images. As well as allowing the image-specific concentration parameters ($c_{cyt}$ and $c_{mem}$) to be learned, this procedure also allows the global signal profiles $s_{mem}$ and $s_{cyt}$ to be optimized and take any arbitrary form, allowing the model to generalize beyond a simple Gaussian PSF model and account for complex sample-specific light-scattering behaviors. We describe this procedure below. In practice, we found that this additional flexibility is necessary to minimize model bias and prevent underfitting (Fig. EV5F).

Analysis was performed in Python using the differentiable programming package JAX (Bradbury et al, 2018). All optimizations were performed with an Adam optimizer (Kingma and Ba, 2015) and a learning rate of 0.005, and run until the loss function (mean squared error) was stabilized.

### Model training
Model training was performed in a two step process (Fig. EV5B). $s_{cyt}$ was trained by performing gradient descent on images of cytoplasmic NeonGreen protein with the $s_{cyt}$ and $c_{cyt}$ as free parameters, and the membrane signal contribution fixed to zero. Training was performed on 17 images in batch with a shared $s_{cyt}$, resulting in the optimized $s_{cyt}$ profile shown in Fig. EV5C.

$s_{mem}$ was trained by performing gradient descent on images of polarized PAR-2 with $s_{mem}$, $c_{mem}$ and $c_{cyt}$ as free parameters, and $s_{cyt}$ fixed to the previously determined profile. The use of polarized images, along with the assumption of a uniform cytoplasmic contribution, allows the model to learn $s_{mem}$ based on the difference in signal between the anterior and posterior halves of the embryo. To test the generalizability of the model, we performed training on separate batches of wild-type PAR-2, PAR-2(L109R), and PAR-2(C56S) images, as well as heterozygous PAR-2 images with a single mNG-tagged copy (50% signal). Training was performed on ten images with a shared $s_{mem}$ for each batch. We found the resulting shape to be similar between all lines (Fig. EV5D). Notably, profiles are asymmetric, with a higher signal at the internal portion of the curve. We reason that this is due to out of focus contributions from membrane protein above and below the plane of the image (Fig. EV5E), which is not accounted for by previous methods. For subsequent quantification steps we used a model trained on the full dataset of 50 images (10 for each condition) (Fig. EV5D, black line).

### Quantification
With $s_{cyt}$ and $s_{mem}$ fixed to the values determined above, quantification was performed on images by performing gradient descent with $s_{cyt}$ and $s_{mem}$ fixed and $c_{cyt}$ (the uniform cytoplasmic concentration) and $c_{mem}$ (an array of membrane concentrations around the embryo) optimized as free parameters.

### Calibration of membrane and cytoplasmic units
$c_{cyt}$ and $c_{mem}$ at this point are in their own arbitrary units, and so a conversion parameter is required to put them into common units. To calibrate this conversion parameter, we quantified the effects on raw membrane and cytoplasm concentration measurements of redistributing a fixed pool of protein from the cytoplasm to the membrane, using an optogenetics PH::eGFP::LOV/ePDZ::mCherry system (Fielmich et al, 2018). Embryos were exposed to blue light for 10 s to promote an interaction between ePDZ and LOV, leading to a rapid uniform recruitment of ePDZ::mCherry to the membrane and a rebalancing of the total protein pool (Fig. EV5G). Expressing the total pool of protein before and after blue light exposure as:

$$T = C + \psi c M \, (\text{before}) \qquad T = C' + \psi c M' \, (\text{after})$$

where $C/C'$ and $M/M'$ are mean membrane and cytoplasmic concentrations in raw model units before/after exposure, the unit conversion factor $c$ can be calculated by comparing the gain in $M$ post-exposure to the loss in $C$:

$$c = \frac{C - C'}{\psi(M' - M)}$$

Full quantification data for wild-type, C56S and L109R PAR-2 in *par-3(wt)* and *par-3(it71)* conditions is shown in Fig. EV1B. Membrane and cytoplasmic concentrations have been converted to equivalent units using the conversion parameter $c$, and all measures

have been normalized to the cytoplasmic concentration of wild-type PAR-2 in *par-3(wt)* conditions. Here and throughout the paper, posterior membrane concentrations are defined as the mean concentration within the posterior-most 20% of the plasma membrane. Posterior M:C ratio is defined as the posterior membrane concentration divided by the (uniform) cytoplasmic concentration. The fraction at the PM is defined as the amount of protein at the plasma membrane divided by the total amount in the cell (cytoplasmic + membrane). Fraction at the posterior PM is the corresponding measure limited to the posterior-most 50% of the PM. Peak membrane concentration in Fig. 4E is defined as the highest concentration within any 20% area of the membrane.

## Scoring cooperativity

We consider a system in which protein exchanges between the cytoplasm (*c*) and membrane (*m*) with the following governing equation:

$$\frac{dm}{dt} = k_{on}c - k_{off}m$$

where $k_{on}$ and $k_{off}$ are membrane binding and unbinding rates. At equilibrium ($\frac{dm}{dt} = 0$), the following condition holds:

$$m = \frac{k_{on}}{k_{off}}c$$

We consider a cooperative system in which $k_{on}$ and/or $k_{off}$ vary as a function of *m*. The precise form will depend on mechanistic details, but, for the purposes of scoring cooperativity, we assume that the quantity $k_{on}/k_{off}$ will take the general form of an exponential ($\beta m^\lambda$) across the relevant range of concentrations. We can then rewrite the equilibrium condition as

$$m = \beta c^\alpha$$

where $\alpha = 1/(1 - \lambda)$. For example, a system in which $k_{on}$ is proportional square-root of membrane concentration (with constant $k_{off}$) will have $\lambda = 0.5, \alpha = 2$, whereas a linear system will have $\lambda = 0, \alpha = 1$. Here, we can see that, for any $\alpha \neq 1$, equilibrium ratios between *m* and *c* will be concentration-dependent.

To score the cooperativity of in vivo systems, we used graded RNAi, along with the image quantification procedure described previously, to quantify *m* and *c* in systems with varying quantities of protein. Then, using a log-transformed version of the equilibrium condition

$$\log_{10}(m) = \alpha \log_{10}(c) + \beta$$

we quantified $\alpha$ and $\beta$ by performing linear regression on the log-transformed data, with $\alpha$ given as the cooperativity score. Probability distributions for $\alpha$ were calculated by bootstrapping.

## RING domain expression and purification

The DNA sequence for PAR-2 residues 40–120 (containing the core RING domain and flanking dimerization helices) was amplified from plasmid pNH46 and cloned into the pETM11 His-Sumo vector (provided by the Crick Structural Biology STP). An L109R

mutant construct was generated by site-directed mutagenesis. Plasmids were verified by sequencing.

Protein was expressed in Rosetta (DE3) cells overnight at 16 °C in LB media supplemented with 50 mM zinc sulfate. The protein was then purified by affinity chromatography (Ni-NTA agarose kit from Qiagen), and the tag was removed by treatment with SenP2 protease (provided by the Crick Structural Biology STP). Protein was further purified by ion exchange chromatography (Cytiva HiTrapMonoQ 1 ml column) and size-exclusion chromatography (Cytiva Superdex 75 increase 10/300 column).

## SEC-MALS

Size-exclusion chromatography coupled multi-angle laser light scattering (SEC-MALS) was used to determine the weight-averaged mass distribution of wild-type PAR-2 and a dimer interface mutant (L109R). Samples ranging from 10 to 0.5 mg/ml were applied in a volume of 100 µl to a SuperdexTM INCREASE 75 10/300 GL column equilibrated in 20 mM Tris-HCl, 150 mM NaCl, 0.5 mM TCEP pH 7.5 at a flow rate of 1.0 ml/min. The scattered light intensity and the protein concentration of the column eluate were recorded using a DAWN-HELEOS laser photometer and OPTI-LAB T-rEX differential refractometer, respectively. The weight-averaged molecular weight ($M_W$) of material contained in chromatographic peaks defined by peak width at half max was determined from the combined data from both detectors using the ASTRA software version 6.0.3 (Wyatt Technology Corp., Santa Barbara, CA, USA). For a system containing a mix of monomers and dimers, this can be modeled as:

$$M_W = \frac{n_m W_m^2 + n_d W_d^2}{n_m W_m + n_d W_d}$$

where $W_m$ and $W_d$ are the molecular weight of monomer and dimer molecules (= 9.23474 and 18.46948 kDa for the PAR-2 RING domain), and $n_m$ and $n_d$ (the number of monomer and dimer molecules in the sample) can be described as a function of total concentration using a dimerization model (see Appendix Supplemental Methods, Additional model details).

## Analytical ultracentrifugation

Sedimentation equilibrium experiments were performed in a Beckman Optima-AUC analytical ultracentrifuge using aluminum double-sector centerpieces in an An-50 Ti rotor. Solvent density and the protein partial specific volumes were determined as described (Laue et al, 1992). Prior to centrifugation, PAR-2 RING samples were dialyzed exhaustively against the buffer blank (20 mM Tris-HCl, 150 mM NaCl, 0.5 mM TCEP). Samples (150 µL) and buffer blanks (160 µL) were loaded into the cells and after centrifugation for 30 h at 20,000 rpm interference data were collected at 2 hourly intervals until no further change in the profiles was observed. The rotor speed was then increased to 24,000 rpm, and the procedure repeated. Data were collected on samples at three different PAR-2 RING concentrations. The program SEDPHAT (Vistica et al, 2004) was used to initially determine weight-averaged molecular masses by nonlinear fitting of individual multi-speed equilibrium profiles to a single-species ideal

solution model. Inspection of these data revealed that the molecular mass showed significant increase over the monomer molecular weight. Therefore, global fitting of the data to a monomer-dimer model incorporating the data from multiple speeds and multiple sample concentrations was applied to extract the monomer-dimer equilibrium dissociation constant ($K_D^{dim}$).

## Structure prediction

The PAR-2 RING structure was predicted using AlphaFold-Multimer (Evans et al, 2022; Jumper et al, 2021) via the online AlphaFold Colab notebook, using a dimer model with residues 40–120 of PAR-2. High prediction confidence (pLDDT >90) was reported for the majority of the output (residues L50 to K117).

## Modeling—equilibrium model

### Four-species thermodynamic model

We consider a system in which protein is both dimerizing and exchanging between membrane and cytoplasmic pools. There are four species to consider: membrane monomers ($m_2$), membrane dimers ($m_2$), cytoplasmic monomers ($c_1$) and cytoplasmic dimers ($c_2$). The membrane is modeled as a thin volume compartment with thickness $a$ equal to the protein diameter (Appendix Fig. S2), with protein exchanging between this compartment and the underlying cytoplasm. Thus, total protein amounts $c_{tot}$ are conserved according to

$$c_{tot} = c_c + a\psi c_m \qquad (1)$$

where $c_c$ ($= c_{c_1} + c_{c_2}$) and $c_m$ ($= c_{m_1} + c_{m_2}$) are protein concentrations in the cytoplasmic and membrane compartments, and $\psi$ is the membrane surface area to cytoplasmic volume ratio. Protein diameter $a$ has not been experimentally determined for PAR-2, but given a molecular weight of 69.95 kDa, the diameter is expected to be ~5 nm (Erickson, 2009). Chemical equilibrium is defined by the following conditions:

$$\mu_c = \mu_m \qquad (2)$$

where $\mu_m$ and $\mu_c$ are effective chemical potentials for the membrane and cytoplasm, which take the following form when we assume dimerization equilibrium (see Supplemental model description for full derivation) (Flory, 1942):

$$\frac{\mu_c}{RT} = \ln\left(\frac{c_c}{c_0}\right) - \frac{1}{2}\ln\left(1 + \frac{4c_c}{K_D^{dim}} + \sqrt{1 + \frac{8c_c}{K_D^{dim}}}\right) + 1 + s_0$$

$$\frac{\mu_m}{RT} = \ln\left(\frac{c_m}{c_0}\right) + \ln\left(K_D^{mem}\right) - \frac{1}{2}\ln\left(1 + \frac{4c_m}{K_D^{dim}} + \sqrt{1 + \frac{8c_m}{K_D^{dim}}}\right) + 1 + s_0$$

where $K_D^{dim}$ is the dimerization dissociation constant, $K_D^{mem}$ is the membrane dissociation constant, $c_0$ is a reference concentration (1 molar), $R$ is the gas constant, $T$ is the temperature in Kelvins and $s_0$ is a reference entropy (note that $R$, $T$, $c_0$ and $s_0$ cancel out in the equilibrium condition). Thus, for a given amount of total protein $c_{tot}$, and given values of the dissociation constants $K_D^{dim}$ and $K_D^{mem}$, equilibrium membrane and cytoplasmic concentrations can be calculated according to the equilibrium condition (Eq. 2) and

conservation law (Eq. 1). Then, within each compartment, monomer and dimer concentrations are given by the following expressions (see Appendix Supplemental Methods. Additional model details for full derivation):

$$c_1 = \frac{K_D^{dim}}{4}\left(\sqrt{1 + \frac{8c}{K_D^{dim}}} - 1\right)$$

$$c_2 = \frac{K_D^{dim}}{4}\left(\frac{4c}{K_D^{dim}} - \sqrt{1 + \frac{8c}{K_D^{dim}}} + 1\right)$$

### Scoring cooperativity

In Fig. 3B, cooperativity was calculated by solving systems at equilibrium with $a = 5$ nm, $\psi = 0.174$ μm$^{-1}$ (Goehring et al, 2011b) and $c_{tot}$ varying from 27 to 0.27 nM, and performing linear regression on log-transformed equilibrium concentrations:

$$\log_{10}(c_m) = \alpha \log_{10}(c_c) + \beta$$

with the slope ($\alpha$) given as the cooperativity score.

### Fitting model to in vivo PAR-2 data

To fit this model to our in vivo PAR-2 rundown data, concentrations were first converted from arbitrary units to a best estimate of absolute concentration. To do so, we made use of previous measurements by Gross et al (2019) that estimated cytoplasmic PAR-2 concentrations in wild-type polarized cells to be 10.4 nM, and normalized our concentration measurements accordingly. Wild-type and L109R data were then fit simultaneously to a model in which $K_D^{mem}$ is shared between wild type and L109R, with $K_D^{dim}$ (L109R) as a free parameter and $K_D^{dim}$ (wt) either free (Fig. 3F) or constrained to the value experimentally determined by AUC (Fig. EV2). By default, we use $a = 5$ nm, however, given uncertainty over the true value of $a$, we performed additional fits with $a = 0.5$ nm and 50 nm for comparison (Appendix Fig. S2).

### Six species thermodynamic model

To include an internal membrane compartment, two additional species were added representing internal membrane-bound monomers ($n_1$) and dimers ($n_2$), leading to the new conservation term

$$c_{tot} = c_c + a(\psi c_m + \phi c_n)$$

where $c_n$ ($= c_{n_1} + c_{n_2}$) is the concentration in the internal membrane compartment, and $\phi$ is the internal membrane surface area to cytoplasmic volume ratio (for simplicity, we assume that $\psi = \phi = 0.087$ μm$^{-1}$ (=0.174/2, reflecting plasma membrane availability in the posterior half)). Equilibrium is given by the new condition:

$$\mu_c = \mu_m = \mu_n$$

with $\mu_n$ given as follows (see Supplemental model description for full derivation):

$$\frac{\mu_n}{RT} = \ln\left(\frac{c_n}{c_0}\right) + \ln\left(K_D^{int}\right) - \frac{1}{2}\ln\left(1 + \frac{4c_n}{K_D^{dim}} + \sqrt{1 + \frac{8c_n}{K_D^{dim}}}\right) + 1 + s_0,$$

where $K_D^{int}$ is the dissociation constant for internal membranes. For our simulations, we used $c_{tot} = 27$ nM, based on previous estimates of the cytoplasmic PAR-2 concentration in polarized wild-type cells

(Gross et al, 2019) and our estimate of the cytoplasmic fraction of PAR-2 in these conditions (Fig. EV1B).

## Modeling—kinetic model

We extend our thermodynamic model using transition state theory to derive the following concentration-dependent on and off rates (see Supplemental model description for full derivation)(Sneppen and Zocchi, 2005; Weber et al, 2019):

$$k_{on} = \frac{\widetilde{\Lambda}}{\sqrt{1 + \frac{4c_c}{K_D^{dim}}} + \sqrt{1 + \frac{8c_c}{K_D^{dim}}}}$$

$$k_{off,m} = \frac{\widetilde{\Lambda} K_D^{mem}}{\sqrt{1 + \frac{4c_m}{K_D^{dim}}} + \sqrt{1 + \frac{8c_m}{K_D^{dim}}}}$$

$$k_{off,n} = \frac{\widetilde{\Lambda} K_D^{int}}{\sqrt{1 + \frac{4c_n}{K_D^{dim}}} + \sqrt{1 + \frac{8c_n}{K_D^{dim}}}}$$

where $c_c$, $c_m$, and $c_n$ are molar concentrations in the cytoplasmic, plasma membrane and internal membrane compartments, and $K_D^{dim}$, $K_D^{mem}$, $K_D^{int}$ are dissociation constants for dimerization, plasma membrane dissociation and internal membrane dissociation as previously defined. $\widetilde{\Lambda}$ is a kinetic pre-factor that scales the rates according to kinetic details (see Supplemental model description for details). We used experimentally determined off-rate measurements to calibrate $\widetilde{\Lambda}$ for PAR-2. FRAP measurements put the plasma membrane unbinding rate in wild-type polarized cells at $0.0073\ s^{-1}$ (Goehring et al, 2011a). With $K_D^{dim} = 358\ nM$ (AUC), $K_D^{mem} = 10^{-2.43}$ (model fit), and $c_m = 47.6\ \mu M$ (our quantification of mean plasma membrane concentration in polarized cells assuming $a = 5\ nm$), this gives $\widetilde{\Lambda} = 46.4\ M\ s^{-1}$. Using these concentration-dependent rate expressions, ODE systems were set up with the following governing equations:

$$\frac{dc_c}{dt} = a[\psi(-k_{on}c_c + k_{off,m}c_m) + \phi(-k_{on}c_c + k_{off,n}c_n)]$$

$$\frac{dc_m}{dt} = k_{on}c_c - k_{off,m}c_m$$

$$\frac{dc_n}{dt} = k_{on}c_c - k_{off,n}c_n$$

Systems were initiated from an equilibrium state with $K_D^{mem} = 1$ and $c_{tot} = 27\ nM$. At time zero, $K_D^{mem}$ was decreased to simulate the onset of posterior plasma membrane availability, and ODE systems were simulated, with $\psi = \phi = 0.087\ \mu m^{-1}$, $a = 5\ nm$ and $\widetilde{\Lambda} = 4\ M\ s^{-1}$.

## Statistics

Measurements were performed on individually dissected embryos, typically obtained across multiple days/experiments to obtain sufficient sample size. Datapoints reflect individual embryos and we show all embryos unless otherwise noted. Probability distributions for statistical metrics, i.e., mean effect size, (Table EV2) were estimated using a bootstrapping method with a total of 10,000 bootstrap samples, with 95% confidence bounds calculated as the 2.5th, and 97.5th percentiles of these probability distributions (Efron and Tibshirani, 1993). Blinding was not used.

## Data availability

Original source data and code are available at https://doi.org/10.25418/crick.24942786. Additional source code and documentation is available at https://github.com/goehringlab.

The source data of this paper are collected in the following database record: biostudies:S-SCDT-10_1038-S44318-024-00123-3.

## Peer review information

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

## Acknowledgements

The authors would like to thank Laura Masino, the Schreiber and Rittinger Labs, and the Structural Biology Scientific Technology Platform at the Crick for help in the purification and analysis of PAR-2, and the Goehring Lab and Katrin Rittinger for comments on the manuscript. Anti-RAB antibodies were a gift from Anne Spang. Some strains were provided by the Caenorhabditis Genome Center (CGC), which is funded by NIH Office of Research Infrastructure Programs (P40 OD010440). This work was supported by the Francis Crick Institute, which receives its core from Cancer Research UK (CC2119, CC2068, CC2029), the UK Medical Research Council (CC2119, CC2068, CC2029), and the Wellcome Trust (CC2119, CC2068, CC2029). DZ and RR gratefully acknowledge funding from the Max Planck Society and the European Union (ERC, EmulSim, 101044662).

## Author contributions

**Tom Bland**: Conceptualization; Resources; Formal analysis; Investigation; Methodology; Writing—original draft; Writing—review and editing. **Nisha Hirani**: Investigation; Methodology; Writing—review and editing. **David C Briggs**: Resources; Formal analysis; Investigation; Methodology; Writing—review and editing. **Riccardo Rossetto**: Formal analysis; Investigation; Methodology; Writing—original draft; Writing—review and editing. **KangBo Ng**: Formal analysis. **Ian A Taylor**: Formal analysis; Funding acquisition; Writing—review and editing. **Neil Q McDonald**: Supervision; Funding acquisition; Writing—review and editing. **David Zwicker**: Formal analysis; Supervision; Funding acquisition; Investigation; Writing—review and editing. **Nathan W Goehring**: Conceptualization; Supervision; Funding acquisition; Investigation; Writing—original draft; Project administration; Writing—review and editing.

Source data underlying figure panels in this paper may have individual authorship assigned. Where available, figure panel/source data authorship is listed in the following database record: biostudies:S-SCDT-10_1038-S44318-024-00123-3.

## Funding

## Disclosure and competing interests statement

The authors declare no competing interests.

# Expanded View Figures

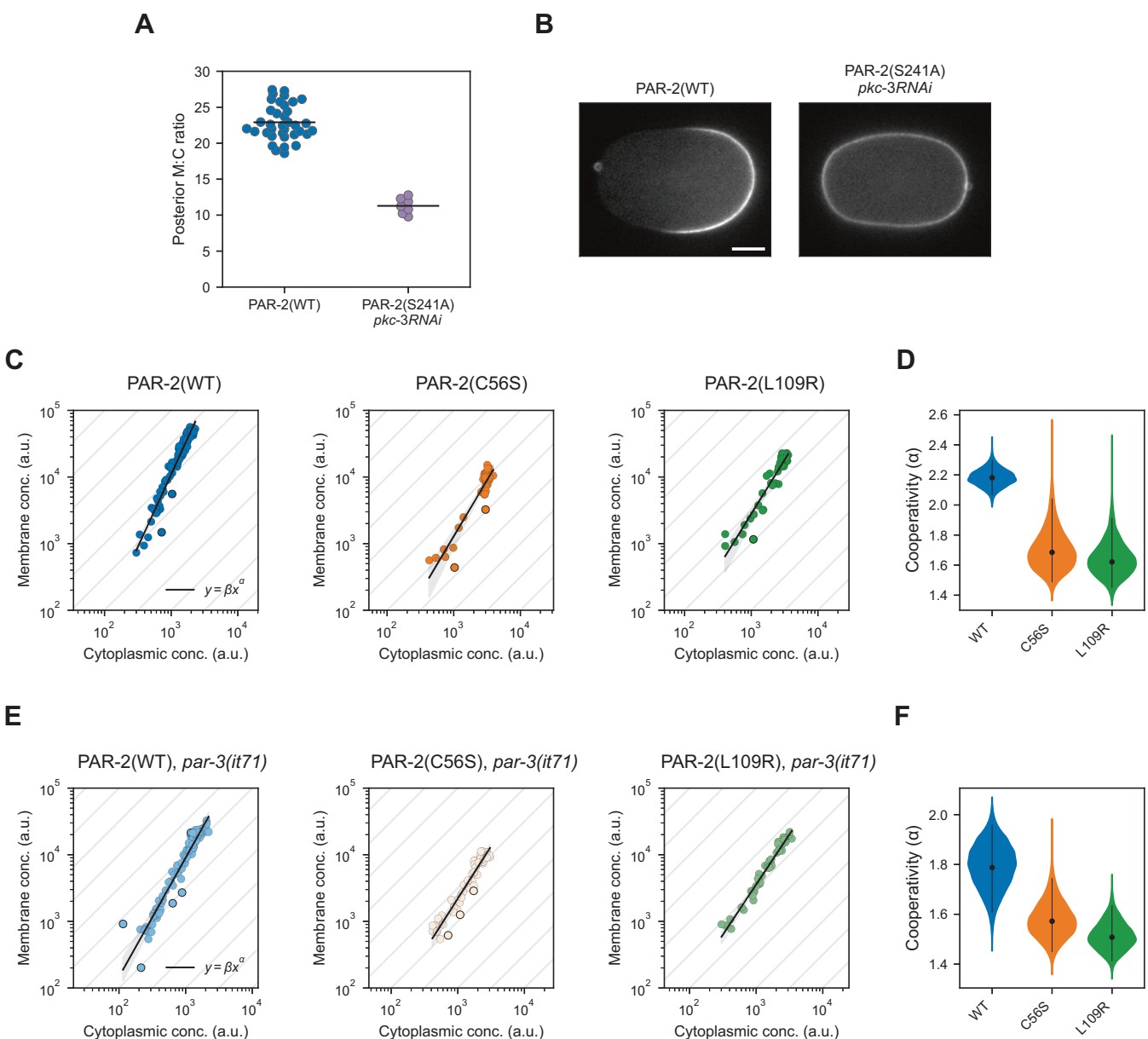

**Figure EV1. PAR-2 cooperativity measurements in datasets stratified by polarity state.**

(**A**) Quantification of posterior membrane-to-cytoplasmic ratio (M:C) ratio in polarized (WT) and uniform (PAR-2(S241A); *pkc-3(RNAi)*) conditions. Similar to Fig. 1E,I, but PAR-2 is rendered uniform by combining the PAR-2(S241A) mutation that disrupts the key PKC-3 phosphorylation site with *pkc-3(RNAi)*. Combining S241A and *pkc-3(RNAi)* was required to achieve reliably uniform PAR-2 distributions in all embryos. Note PAR-2(WT) data is reproduced from (1E) for comparison. (**B**) SAIBR-corrected images of mNG::PAR-2 in polarized (WT) and uniform (PAR-2(S241A); *pkc-3(RNAi)*) conditions. (**C**) Plots of membrane vs cytoplasmic concentrations of PAR-2 and PAR-2 RING mutants (C56S, L109R) in polarized, *par-3(WT)* cells. (**D**) Probability distribution of cooperativity scores determined from *par-3(WT)* data. (**E**) Plots of membrane vs cytoplasmic concentrations of PAR-2 and PAR-2 RING mutants (C56S, L109R) in unpolarized, *par-3(it71)* cells. (**F**) Probability distribution of cooperativity scores determined from *par-3(it71)* data. Data information: In (**A**), datapoints represent individual embryos. All data shown, mean indicated. (**B**) Scale bar = 10 μm. (**C, E**) Black lines show fits to a linear regression model with 95% confidence bands calculated by bootstrapping shown. (**D, F**) Best fits to the full dataset (dots) are shown with probability distributions of cooperativity (violin plot) and 95% confidence intervals (lines) calculated by bootstrapping. Additional statistics are available in Table EV2.

## A

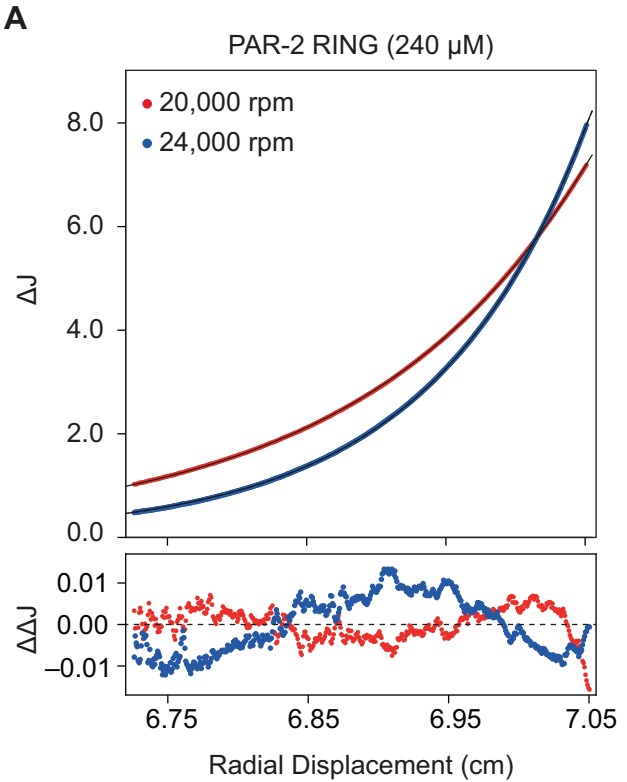

PAR-2 RING (240 µM)

- 20,000 rpm
- 24,000 rpm

## B

**Hydrodynamic parameters**

| Protein | PAR-2 RING |
|---|---|
| [a]v (mL.g$^{-1}$) | 0.728 |
| [b]ρ (g.mL$^{-1}$) | 1.005 |
| [c]η (x10$^2$) (g$^{-1}$ cm$^{-1}$ s$^{-1}$) | 1.022 |
| [d]M$_r$ | 9,235 |
| [e]ε$_{280}$ (M$^{-1}$ cm$^{-1}$) | 1,300 |
| [f]J$_{inc}$ (M$^{-1}$.cm$^{-1}$) | 25,396 |

[a]Protein partial specific volume; [b]Buffer density; [c]Buffer viscosity [d]Molar mass calculated from the protein sequence; [e]Molar absorbance extinction coefficient; [f]Molar fringe increment.

**Sedimentation equilibrium**

**PAR-2 RING**

| C (µM) | 85 | 125 | 240 | 85-240 |
|---|---|---|---|---|
| [a]M$_w$ (kDa) | 19.2 | 18.8 | 18.0 | 18-19.2 |
| [b]K$_D^{dim}$ (µM) | 0.36 | 0.37 | 0.33 | 0.36 |
| [c]rmsd | 0.006 | 0.006 | 0.004 | 0.004 – 0.006 |
| [d]χ$^2$ | | | | 1.25 |

[a]weight averaged molecular weight derived from Global analysis of individual samples using single species model; [b]monomer-dimer equilibrium dissociation constant determined from a global fit using three concentrations and two speeds to a monomer-dimer self-association model; [c]rms deviation observed for each multi-speed sample when fitted individually and globally; [d]global reduced chi-squared from combined fitting of all multispeed data.

---

**Figure EV2. PAR-2 RING self-associates in solution.**

(A) Multi-speed sedimentation equilibrium profiles determined from interference data collected on PAR-2 RING at 240 µM. Data was recorded at the speeds indicated. The solid black lines represent the global best fit to the data (red, blue points) using a monomer-dimer model ($K_D^{dim}$ = 0.36 µM, reduced $\chi^2$ = 1.25). The lower panel shows the residuals to the fit. (B) Full PAR-2 RING sedimentation equilibrium data for AUC performed at multiple concentrations.

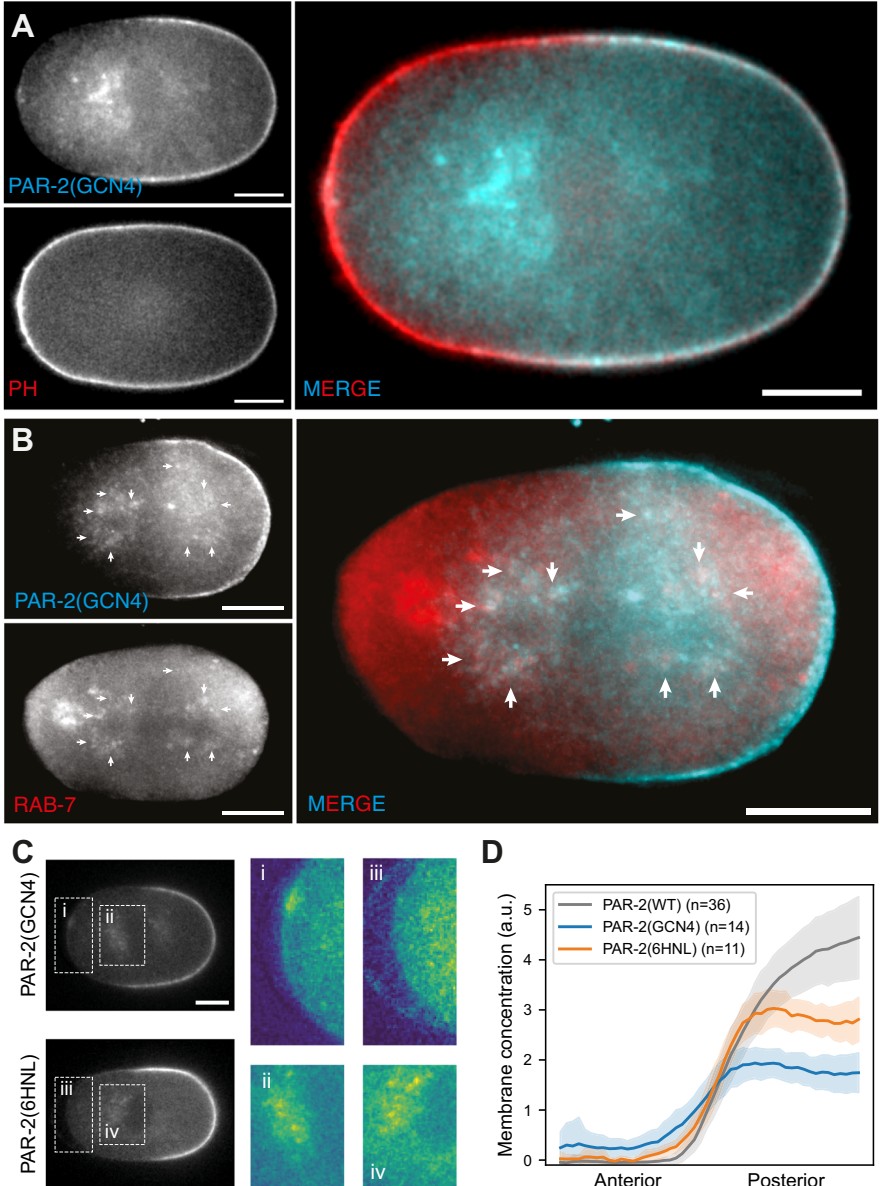

**Figure EV3.  GCN4-dependent dimerization of PAR-2 leads to recruitment to endosomal membranes, which is recapitulated by an alternative dimerization domain.**

(A) Midsection confocal images showing no colocalization between mNG::PAR-2(GCN4) and the plasma membrane marker mCherry::PH$_{PLC\delta1}$. Single channel and merged images shown. Typical embryo shown ($n = 4$). (B). Colocalization of PAR-2(GCN4) with RAB-7 in fixed embryos. Single channel and merged images shown. Arrows highlight sample regions with significant overlap. Images are maximum Z-projections of central 10 × 0.25 μm sections. Typical embryo shown ($n = 4$). (C) SAIBR-corrected images comparing PAR- 2(GCN4) and a version of PAR-2 dimerized via an alternative dimerization domain, 6HNL. Note that PAR-2(6HNL) exhibits similar accumulation on internal membranes and residual signal at anterior membrane. (D) Anterior to posterior membrane concentration profiles of PAR-2(GCN4) and PAR-2(6HNL), with PAR-2(WT) shown for reference. Data Information: Scale bars in (A–C) = 10 μM. (D) Mean ± SD, with number of embryos indicated ($n$).

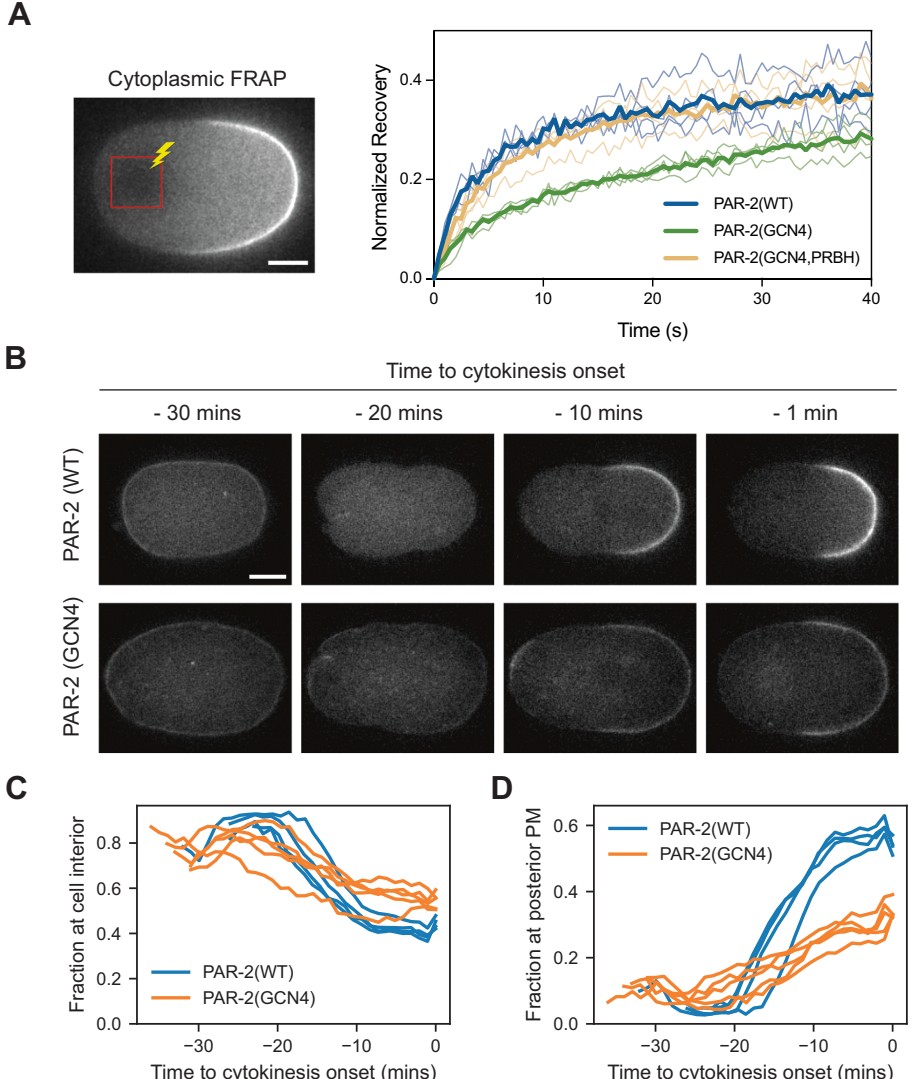

**Figure EV4. PAR-2(GCN4) displays reduced dynamics compared to wild type.**

(A) Normalized recovery curves for cytoplasmic FRAP (see "Methods"). Note that recovery kinetics are reduced for PAR-2(GCN4) compared to wild type, and are restored to near wild-type behavior by mutating residues in the PRBH domain (PRBH). (B) Midplane confocal images of PAR-2(WT) and PAR-2(GCN4) localization from meiosis to cytokinesis onset. Together with quantifications in (C, D), these data show that redistribution from the cell interior to plasma membrane is slowed for PAR-2(GCN4). (C) Quantification of total fraction of PAR-2(WT vs GCN4) in the cell interior over time. (D) Quantification of total fraction of PAR-2(WT vs GCN4) at the posterior plasma membrane over time. Data information: (A) Individual and mean shown. (C, D) Traces from individual embryos shown. Scale bars in (A, B) = 10 μm.

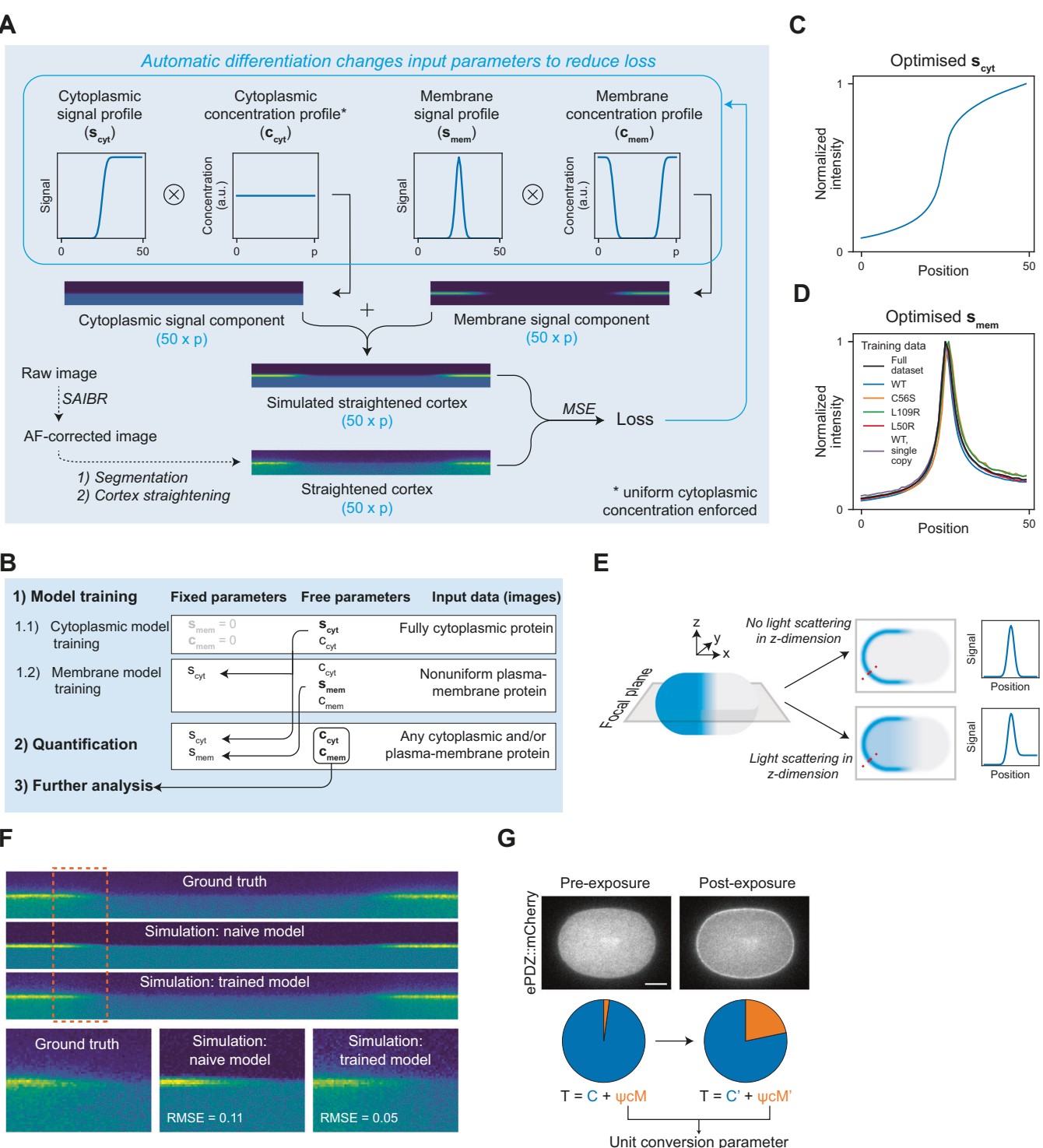

◀ **Figure EV5.   A machine-learning method for extraction of normalized membrane and cytoplasmic protein concentrations from midplane confocal images.**

(A) Schematic of differentiable model for image quantification. See "Methods" for details. (B) Outline of model training and quantification protocol. See "Methods" for details. (C) Cytoplasmic signal profile determined by cytoplasmic model training on images of cytoplasmic mNG. (D) Membrane signal profiles determined by membrane model training on images of wild-type PAR-2, mutant alleles and single-mNG heterozygotes. Black line shows a model trained on the full dataset. (E) Schematic of the expected effects of 3D light scattering on observed midplane signal distributions from membrane protein. (F) Example of ground truth (SAIBR-corrected) and simulated images for an mNG::PAR-2(L109R) embryo. Naive model refers to a model in which membrane and cytoplasmic signal profiles are fixed to a Gaussian and error function. Trained model refers to a model in which cytoplasmic and membrane profiles have been trained according to the process outlined in (B). Gaussian noise has been added to simulated images to allow for closer visual comparison to the ground truth image. RMSE: root mean square error. (G) Optogenetics system used to calibrate cytoplasmic and membrane concentration units. Exposure to blue light promotes an interaction between ePDZ::mCherry and membrane-tethered PH::eGFP::LOV, causing recruitment of ePDZ::mCherry to the membrane. Pie charts show the amount of total ePDZ::mCherry in the cytoplasm, C, and membrane, M, before and after exposure to blue light, which sum to a constant value T. A unit conversion factor (*c*) can be calculated by solving the equations shown, with ψ being the surface:volume ratio. Scale bar = 10 μm.

