## [Peer Review File · The EMBO Journal]

Optimized PAR-2 RING dimerization mediates cooperative and selective membrane binding for robust cell polarity

Tom Bland, Nisha Hirani, David Briggs, Riccardo Rossetto, KangBo Ng, Ian Taylor, Neil McDonald, David Zwicker, and Nathan Goehring

Corresponding author: Nathan Goehring (nate.goehring@crick.ac.uk)

Review Timeline:

Submission Date:	4th Sep 23
Editorial Decision:	23rd Oct 23
Revision Received:	21st Feb 24
Editorial Decision:	22nd Mar 24
Revision Received:	1st May 24
Accepted:	8th May 24

Editor: Ieva Gailite

Transaction Report:

Dear Nate,

Thank you for providing a preliminary revision plan for your manuscript. I have now discussed it with reviewers #2 and #3, and, based on their input, I would like to invite a revised manuscript broadly as indicated in your revision plan. I have also included the comments by referee #3 below.

When revising the manuscript, please provide full information on the number of experimental replicates and used statistical analysis to aid the assessment of the manuscript. Referee #2 also asks to clarify the use of different scales and methods in different figures. Based on the referee input, please also rewrite the text to clarify the aims of the study for the broader audience and to distinguish between PAR2-mediated polarity establishment and maintenance. To this purpose, referee #3 suggests using the term "PAR2 domain formation" to avoid ambiguity. In their assessment of the revision plan, reviewer #2 in particular points out that they have remaining concerns regarding the PAR2 RING domain dimerisation experiments, and more convincing analysis is needed - please consider if you would be able to address this with alternative approaches.

We generally allow three months as standard revision time. As a matter of policy, competing manuscripts published during this period will not negatively impact on our assessment of the conceptual advance presented by your study. However, please contact me as soon as possible upon publication of any related work to discuss the appropriate course of action. Should you foresee a problem in meeting this three-month deadline, please let us know in advance in order to arrange an extension.

When preparing your letter of response to the referees' comments, please bear in mind that this will form part of the Review Process File and will therefore be available online to the community. For more details on our Transparent Editorial Process, please visit our website: <https://www.embopress.org/page/journal/14602075/authorguide#transparentprocess>. Please also see the attached instructions for further guidelines on preparation of the revised manuscript.

I would be happy to discuss the revision further, e.g., via a Zoom call. Thank you for the opportunity to consider your work for publication, and I look forward to your revision.

With best wishes,

Ieva

- a point-by-point response to the referees' comments, with a detailed description of the changes made (as a word file).
- a word file of the manuscript text.
- individual production quality figure files (one file per figure)
- a complete author checklist, which you can download from our author guidelines

(<https://www.embopress.org/page/journal/14602075/authorguide>).
- Expanded View files (replacing Supplementary Information)
Please see out instructions to authors
<https://www.embopress.org/page/journal/14602075/authorguide#expandedview>

We realize that it is difficult to revise to a specific deadline. In the interest of protecting the conceptual advance provided by the work, we recommend a revision within 3 months (21st Jan 2024). Please discuss the revision progress ahead of this time with the editor if you require more time to complete the revisions.

Referee #1:

Cell polarity plays a fundamental role in determining the fate, shape, and function of tissues and organs during development. The establishment of cell polarity has been extensively studied with the *C. elegans* zygote, which polarises through segregation of PAR proteins right after fertilisation. Bland et al. investigated the non-linear positive feedback system of PAR-2 polarity and demonstrated that proper level of oligomerisation through the RING domain is essential for PAR-2 to establish correct size and cortical concentration. The RING domain of PAR-2 has been shown to stabilise it at the cell cortex, probably acting against cortical exclusion by PKC-3. This study demonstrated that the cortically-targeted RING domain (but not the cytoplasmic RING domain) can be trapped at the posterior cortex by endogenous PAR-2. Manipulating PAR-2's dimerisation activity by introducing either L109R or L50R mutations and inserting the GCN4 dimerisation tag compromised proper polarisation of PAR-2 and caused embryos hypersensitive to the loss of cortical flow, another symmetry-breaking cue for the zygote. Their mathematical modelling also supports the role of dimerisation in the non-linear feedback of PAR-2. Overall, their model is sufficiently supported by multiple lines of evidences and will attract general readers in cell, development, and molecular biology area. The manuscript is well written with a streamlined logic and reasonable discussions. I would enthusiastically support its publication in EMBO journal, once the authors address minor points below.

To distribute PAR-2 throughout the cortex in Fig. 1, the authors also need to check *pkc-3(RNAi)* condition, because *par-3(it71)* may allow PKC-3 to phosphorylate PAR-2 and reduce its cortical concentration. This simple experiment will further strengthen the co-operative feedback model.

They should show whether the size of cortical PAR-2(L109R) and PAR-2(L50R) domain could be smaller than that of PAR-2(wild type). Show the images of GFP-tagged PAR-2(L109R) and PAR-2(L50R) in zygotes.

The identity of the cytoplasmic distribution of PAR-2(GCN4 and 6HNL) is totally unclear. They may want to check if PAR-2(GCN4 and 6HNL) could co-localise with endosomal membrane markers such as RAB-5/RAB-7/RAB-11.

Enhanced dimerisation of PAR-2 may not affect its specificity to lipids but would simply retard the exchange of PAR-2 with the plasma membrane lipids. Previous study has shown that the plasma membrane marker PHPLC undergoes internalisation from the cortex to the cytoplasm in a manner dependent on microtubules (Redemann et al., 2010). Thereby, it is possible that the cytoplasmic distribution of PAR-2(GCN4 and 6HNL) may be a consequence of hyper-stable association of PAR-2 to the internalised plasma membrane lipids. The authors may want to test this possibility by FRAP analysis of NG::PAR-2(GCN4 and 6HNL) at the cell cortex.

I would recommend the authors to avoid "data not shown" in the main text. Additional data can be shown in Supplementary Figures.

Previous study by Arata et al. (2016) used single-molecule imaging techniques and observed the formation of trimeric and tetrameric forms of GFP::PAR-2 at the cell cortex. Bland et al. observed predominant dimerisation of the RING domain through *in vitro* experiments. I would appreciate if the authors discuss such discrepancy in the main text. Statistical comparison should be done for Figs. 1E, 1G, 2I-K, 5F, 5H, S2D, S2H, and S4B.

Method section should include a "statistics and reproducibility" section, which describes all statistical tests, the numbers of

biological samples and replicates, and data collection and analysis.

Referee #2:

In this work the authors ask how polarity networks can ensure stable polarization while remaining responsive to changing polarity cues or cellular states. They use the *C. elegans* embryo as a model of cell polarity and combine theory and experiments to show that dimerization of the polarity protein PAR-2 via the RING domain contributes to polarity robustness and responsiveness. The data show that both reduction and increased ability of PAR-2 to dimerize compromise the robustness of polarity and the authors suggest that "oligomerization kinetics can be a strategy to optimize dynamic and cooperative intracellular targeting".

Major comments

1) My first major comment is that oligomerization of PAR-2 was already shown to be important in cell polarity establishment (ARATA et al, Developmental cell, 2016) and the authors themselves (Illukkumbura et al) have shown that clustering of PAR proteins at the membrane increases the stability of membrane association (and the authors also mention this in the discussion about the present work). Here the authors suggest that optimized dimerization of PAR-2 (not too strong, not too weak) is important for cooperative membrane recruitment. However, how they distinguish, experimentally, recruitment from decreased dissociation of PAR-2? A possibility is that the dimerization or oligomerization at the plasma membrane reduces the dissociation rate of PAR-2, which is what Arata et al already showed, what the authors suggested in Illukkumbura et al for other PAR proteins and they conclude in the discussion of this work. Basically, I am not sure what is the point the authors want to stress here: is it cooperative recruitment or reduced dissociation?.

2) Figure 1: I disagree with the first statement of the result section on how the binding of PAR-2 is regulated (mass action versus cooperativity). What happens if the cortical PAR-2 intensity is normalized to the length of the domain that it occupies? Also, if there is cooperativity in control, why the cooperativity is abolished in par-3 mutants (where the domain occupied by PAR-2 is actually longer)?.

In this same figure, the authors use the C56S mutant of PAR-2 but how this mutant influences membrane localization of PAR-2 is not clear. Is this mutant also interfering with PAR-2 dimerization? If yes, why the authors use other mutants later? How do the results compare to the L109R mutant (which is supposed to disrupt PAR-2 dimerization)? Again, as mentioned above, what are the data that support a feedback mechanism?

3) In the description of figure 2, the authors conclude: "Thus the RING domain appears to be sufficient to mediate recruitment by endogenous PAR-2 provided that it is stabilized at the plasma membrane". I disagree with this conclusion: the domain alone cannot go to the plasma membrane at all. It can go only when fused to a membrane targeting domain, which means it is not sufficient.

4) In figure 2, the authors show dimerization of PAR-2 and the reduction of dimerization of the L109R. There is a major problem with this figure: in the text the authors say that the RING domain of PAR-2 shows a "concentration dependent dimerization shifting from mostly monomeric to mostly dimeric". Now, one of the lowest concentrations used in the wild type PAR-2 RING domain is 0.75 mg/ml (Figure 2F) and this is mostly monomeric, as stated by the authors. To show that dimerization is lost in the L109R mutant and that this is monomeric, the authors used 0.75 mg/ml. At this concentration PAR-2 is already monomeric when wild type and indeed the difference compared to wild type in 2H is minimal. Why only the lowest concentration was used? The authors should use up to 10 mg/ml and show that dimerization is indeed affected in the mutant. In addition, are the values in the axis in figure 2G correct?

As mentioned above, it would be interesting to test the C56S mutant in vitro and see if it influences dimerization.

5) The total level of the different PAR-2 mutants/fusions needs to be measured and compared with the wild type, either by microscopy or by western blot. If the total level can be obtained by summing up the plasma membrane and cytoplasmic values in Table S2, this means that the mutants used (both C56S and L109R) both result in reduced total levels of PAR-2, which makes the interpretation of all the experiments harder and also is not consistent with the paragraph suggesting that the C56S mutant may have stronger phenotypes because of destabilization of the RING domain and lower protein amounts as the L109R seems to do the same.

6) Statistics are missing in most if not all figures. As one example in figure 5H the authors say that the fraction of PAR-2(GCN4, PRBH) at the posterior PM is higher compared to PAR-2(GCN4) but the variability for one of the two is quite high and the difference in value is quite small. Is this significant? If the authors use the same scale in both 5F and H, PAR-2(GCN4) in 5H appears to have approximately the same value as the PAR-2(GCN4-PRBH) in 5H (a bit higher than 0.4), which is not consistent with the argument of the authors.

7) In the plots, does each dot correspond to 1 embryo? It is hard to see how many embryos have been measured which should be indicated in the figure or figure legend together with the number of independent experiments performed.

8) The major concepts are often expressed in a convolute manner which does not help the reader to understand. As a simple

example, in page 7, the authors say: "Varying membrane binding.....". However, they actually vary the K(D), not the K(ON). In the same sentence, the authors say: "for high membrane binding affinity" but they actually take a value that is in the middle (Fig. 3B). Why do they say "high", especially if compared to the value that they take for dimer affinity, which they call intermediate?

9) To increase dimer affinity, the authors create a constitutive PAR-2 dimer by adding the GCN4 leucine zipper motif. One problem of this experiment is that PAR-2GCN4 goes to internal membranes. Additional controls are needed for this experiment. First, an endogenous PAR-2GCN4 fusion with no other tag would be a good control that the localization is indeed altered (in fixed samples by staining). The open question is also: what are these internal membranes? And why only those? Why this dimer does not go, for example, to the ER or the Golgi? (assuming that these internal membranes are not ER or Golgi). So, the selectivity is not totally disrupted as the dimer does not seem to go to all internal membranes but only to a subset of those. What are these membranes? Colocalization with markers is needed to identify the membranes. Have the authors considered the possibility that the reason why PAR-2(GCN4) is less at the plasma membrane is not because the dimerization strength is increase but because PAR-2 is unable to form larger oligomers that would be stabilized at the plasma membrane?

10) The fact that PAR-2(GCN4) in par-3 mutant embryos occupies less internal membranes suggest that this behavior of PAR-2(GCN4) (binding to internal members) is specific to embryos and not oocytes/germline. If this is the case, what is the explanation? In Figure S8B, it seems that PAR-2(GCN4) remains longer at the plasma membrane (picture -20 minutes). Is this the case? And if yes, how to the authors reconcile with their hypothesis? It does not look like here PAR-2 is trapped already in internal membranes. Finally, in figure 5E, the PAR-2GCN4 domain is indeed less intense in wild type but extends more anteriorly.

Minor comments

In the abstracts the authors mention "ubiquitin independent function" of the RING domain. As there are no data about ubiquitin, the sentence should be removed.

At page 3, in the second paragraph, the authors say: "...and which are thought to be targeted for phosphorylation...": why are thought? Are the authors not convinced by the published literature?

The numbers in the PAR-2 scheme in Figure 1B are not clear as they do not correspond to the domains mentioned.

Can the authors show images of the L109R embryos?

To reduce the flows, the authors use nop-1 in one experiment and mlc-4 in another. Any reason for this? Why not using nop-1 in both cases?

In the scheme in figure 1H is practically impossible to see a reduced membrane PAR-2.

Describing the result, the authors state: "We found that PAR-2(GCN4) was enriched at the posterior as PAR-2(WT)". The statement is incorrect as the levels of the PAR-2 fusion at the posterior are much lower than the WT, as stated just below in the text.

Figure 4: for consistency, the authors should show the MC measurements as in figure 1 and 2.

It would be useful if the authors put in the PAR-2 scheme where the GCN4 domain is introduced

When the authors introduce the experiment where they mutate the serines in the PRBH domain, it would be good if they add a sentence introducing the rationale of the experiment.

Referee #3:

Bland et al. present very interesting research addressing a long-standing question in the field of PAR polarity, how PAR-2 pathway can induce zygote polarization (symmetry breaking) in the absence of the actomyosin flow-induced pathway. The problem had been formulated a while ago, mostly in the pioneering works from the lab of Geraldine Seydoux, which are duly

cited. In those papers, it had been shown that self-interaction of PAR-2 via the RING domain plays an important role. Here, the authors further develop this theme and show the crucial contribution of the RING domain dimerization. From the point of view of biological experiment, the manuscript presents careful, meticulous analysis with thoughtful controls and excellent image analysis. I would like to commend the authors on the sheer volume of work, creative use of mutants and attention to detail. This work leads the authors to conclude that the ring domain dimerization introduces positive cooperativity into the binding of PAR-2 to the membrane.

My major issue is with the presented theoretical explanation. The authors take a highly non-standard approach, developing a novel thermodynamics-based theory, completely avoiding the use of routine reaction-diffusion formalism. The biological question is how PAR-2 subsystem can explain cellular polarization - i.e., how this mechanism can induce symmetry breaking. However, this very central question is not addressed by the proposed theory! The authors must provide such an analysis and clearly demonstrate that their simple mechanism can, indeed, break symmetry, e.g., by using linear stability analysis of the homogeneous state of the cell. If the authors claim that this is the case, then we have another problem of fundamental physics nature. In this case, the claim is that any protein that can reversibly dimerize and bind to the membrane is sufficient to break spatial symmetry! This is too big a claim and, frankly, sounds too good to be true. As the system consists of only reversible reactions without any input of energy, i.e., purely equilibrium system, then how does it break symmetry? This seems to contradict the results typically associated with the seminal work of Ilya Prigogine. The authors have to squarely address this puzzling conundrum in both results and discussion and provide clear evidence and analysis.

Referee #3 comments on the revision plan:

I would like to thank the authors for their detailed reply to my criticism. I do appreciate that they do not clearly claim that they provide a mechanism for "symmetry breaking". However, they do use word "polarization" frequently in the body in conjunction with the PAR2 domain formation. The term "polarization" is for many people equivalent to "symmetry breaking". I strongly recommend using instead "PAR2 domain formation" or equivalent to prevent any ambiguity. Further to my early suggestions and taking into account the authors' reply, here are my recommendations:

1. As the lead author is known for his contribution to the "flow pathway", while current work mainly concerns with the "PAR2 pathway" with which other labs are closely associated, in the Introduction, it would be very useful to add a paragraph succinctly recounting the event sequence happening during the zygote polarization. Too many people know something about PAR polarity complexes but very few have a clear view of normal and "alternative" pathways of *C. elegans* zygote polarization. It would be beneficial to highlight that the primary symmetry-breaking event is the entry of the sperm, causative of the local cortex weakening and the following "flow pathway". Offering the readers a few modern reviews, e.g., Kapoor and Kotak PubMed 32597472, would be helpful.
2. In the Discussion, the authors should clearly indicate where exactly in the sequence of events throughout normal and "alternative" polarity establishment they see the role of PAR2 binding cooperativity. This is greatly important to put the results of the paper in the perspective and avoid any potential misunderstanding

Response to Referee Reports (13 February 2024)

We thank the reviewers for careful evaluation of our manuscript and suggestions for improvement. We have addressed all concerns raised as detailed below.

Referee #1:

Cell polarity plays a fundamental role in determining the fate, shape, and function of tissues and organs during development. The establishment of cell polarity has been extensively studied with the *C. elegans* zygote, which polarises through segregation of PAR proteins right after fertilisation. Bland et al. investigated the non-linear positive feedback system of PAR-2 polarity and demonstrated that proper level of oligomerisation through the RING domain is essential for PAR-2 to establish correct size and cortical concentration. The RING domain of PAR-2 has been shown to stabilise it at the cell cortex, probably acting against cortical exclusion by PKC-3. This study demonstrated that the cortically-targeted RING domain (but not the cytoplasmic RING domain) can be trapped at the posterior cortex by endogenous PAR-2. Manipulating PAR-2's dimerisation activity by introducing either L109R or L50R mutations and inserting the GCN4 dimerisation tag compromised proper polarisation of PAR-2 and caused embryos hypersensitive to the loss of cortical flow, another symmetry-breaking cue for the zygote. Their mathematical modelling also supports the role of dimerisation in the non-linear feedback of PAR-2. Overall, their model is sufficiently supported by multiple lines of evidences and will attract general readers in cell, development, and molecular biology area. The manuscript is well written with a streamlined logic and reasonable discussions. I would enthusiastically support its publication in EMBO journal, once the authors address minor points below.

To distribute PAR-2 throughout the cortex in Fig. 1, the authors also need to check *pkc-3(RNAi)* condition, because *par-3(it71)* may allow PKC-3 to phosphorylate PAR-2 and reduce its cortical concentration. This simple experiment will further strengthen the co-operative feedback model.

We have repeated this analysis on *mNG::PAR-2(S241A)*, *pkc-3(RNAi)* embryos, which yielded nearly identical results as *par-3(it71)*. PAR-2(S241A) is resistant to phosphorylation by PKC-3 and in our hands more fully compromises PKC-3 activity against PAR-2 compared to *pkc-3(RNAi)* alone. Note, we include *pkc-3(RNAi)* to reduce cortical flow, which can lead to modest anterior enrichment of PAR-2(S241)(see Illukkumbura et al 2023). New Figure EV1A.

They should show whether the size of cortical PAR-2(L109R) and PAR-2(L50R) domain could be smaller than that of PAR-2(wild type). Show the images of GFP-tagged PAR-2(L109R) and PAR-2(L50R) in zygotes.

Sample images have now been added (Appendix Figure S1).

The identity of the cytoplasmic distribution of PAR-2(GCN4 and 6HNL) is totally unclear. They may want to check if PAR-2(GCN4 and 6HNL) could co-localise with endosomal membrane markers such as RAB-5/RAB-7/RAB-11.

We now include imaging data comparing the localization of PAR-2(GCN4) to RAB-5/RAB-7/RAB-11. Note that we see substantial, though incomplete, overlap with RAB-7, less so for RAB-5/11, suggesting a more general endosomal localization as opposed to enrichment in a specific compartment. See Figure EV3, Appendix Figures S3/S4.

Enhanced dimerisation of PAR-2 may not affect its specificity to lipids but would simply retard the exchange of PAR-2 with the plasma membrane lipids. Previous study has shown that the plasma membrane marker PHPLC undergoes internalisation from the cortex to the cytoplasm in a manner dependent on microtubules (Redemann et al., 2010). Thereby, it is possible that the cytoplasmic distribution of PAR-2(GCN4 and 6HNL) may be a consequence of hyper-stable association of PAR-2 to the internalised plasma membrane lipids. The authors may want to test this possibility by FRAP analysis of *NG::PAR-2(GCN4 and 6HNL)* at the cell cortex.

As noted, the model clearly predicts stronger association of PAR-2(GCN4) with the plasma membrane and FRAP experiments are consistent with this (see Illukkumbura et al 2023). However, we rarely observe PH-PLCd1 positive plasma membrane accumulations in wild-type embryos as they primarily result from MT-pulling of the PM when the actin cortex is defective, suggesting these are not PM internalizations. To address this specifically, we have now generated a dual labeled *mNG::PAR-2(GCN4)*; *mCherry::PH-PLCd1* line and see no recruitment of PH-PLCd1 to internal *mNG::PAR-2(GCN4)* accumulations (Figure EV3), which together with our RAB colocalization analysis suggests these are endosomal, not PM, in nature.

Published data suggest that PAR-2 engages in charge-based recognition of membranes via basic hydrophobic (BH) domains and as noted in the text, plasma membrane selectivity can be explained in such contexts by the overall

charge of the respective membrane compartments. Because of the preponderance of PS and PIP2 in the PM, it is the most negatively charged, followed by endosomes, which would explain why we see localization to Rab-labeled compartments in PAR-2(GCN4). A protein that recognizes specific PM-enriched lipids rather than charge would not be expected to show such behavior. For example, the PH domain of PLCd1 (PHPLCd1) specifically binds PIP2 and at least in mammalian cells, tandem PHPLCd1 probes retain their PM specificity - note Hammond et al 2009 (10.1083/jcb.200809073) Figure 4A shows increase in M:C ratios, opposite what we see for PAR-2(GCN4).

That said, as we note in the Discussion, it seems likely that there would be a constant flux of PAR-2 into endosomes due to internalization, unless specifically excluded. While PAR-2(WT) would be able to quickly sense the change in membrane charge that occurs during internalization and thus redistribute back to the PM, PAR-2(GCN4) would be slow to redistribute back to the PM, leading to internal accumulation. This is perhaps why we can't fully eliminate this internal pool in *par-3(it71)* embryos.

I would recommend the authors to avoid "data not shown" in the main text. Additional data can be shown in Supplementary Figures.

As it is peripheral to the study, we have removed all references to ubiquitin-independence.

Previous study by Arata et al. (2016) used single-molecule imaging techniques and observed the formation of trimeric and tetrameric forms of GFP::PAR-2 at the cell cortex. Bland et al. observed predominant dimerisation of the RING domain through in vitro experiments. I would appreciate if the authors discuss such discrepancy in the main text.

We have added text to the Discussion to note the results in Arata et al (2016) and discuss the differences in the two works. We have not included any single particle tracking data in this work and so cannot directly compare our analysis pipelines. A key reason for this is that we believe that single particle tracking is unlikely to accurately report oligomer size given that sub-stoichiometric labeling of PAR-2 (i.e. <10% of particles fluorescent) is required to obtain particle densities low enough to enable accurate tracking and to exclude overlapping particles from confounding measurements (see particle densities in Movies S1-S3 in Arata et al (2016) for example of particle densities used for tracking).

Statistical comparison should be done for Figs. 1E, 1G, 2I-K, 5F, 5H, S2D, S2H, and S4B.

Method section should include a "statistics and reproducibility" section, which describes all statistical tests, the numbers of biological samples and replicates, and data collection and analysis.

We have added the requested "Statistics" section to the Methods and provide a Supplemental Data file (Table EV3) detailing sample numbers, relevant effect sizes and associated 95% confidence intervals for all relevant comparisons.

Referee #2:

In this work the authors ask how polarity networks can ensure stable polarization while remaining responsive to changing polarity cues or cellular states. They use the *C. elegans* embryo as a model of cell polarity and combine theory and experiments to show that dimerization of the polarity protein PAR-2 via the RING domain contributes to polarity robustness and responsiveness. The data show that both reduction and increased ability of PAR-2 to dimerize compromise the robustness of polarity and the authors suggest that "oligomerization kinetics can be a strategy to optimize dynamic and cooperative intracellular targeting".

Major comments

1) My first major comment is that oligomerization of PAR-2 was already shown to be important in cell polarity establishment (ARATA et al, Developmental cell, 2016) and the authors themselves (Illukkumbura et al) have shown that clustering of PAR proteins at the membrane increases the stability of membrane association (and the authors also mention this in the discussion about the present work). Here the authors suggest that optimized dimerization of PAR-2 (not too strong, not too weak) is important for cooperative membrane recruitment. However, how they distinguish, experimentally, recruitment from decreased dissociation of PAR-2? A possibility is that the dimerization or oligomerization at the plasma membrane reduces the dissociation rate of PAR-2, which is what Arata et al already showed, what the authors suggested in Illukkumbura et al for other PAR proteins and they conclude in the discussion of this work. Basically, I am not sure what is the point the authors want to stress here: is it cooperative recruitment or reduced dissociation?.

We have revised our terminology to make our conclusions more clear. We generally used “cooperative recruitment” to describe the cooperative, nonlinear relationship between membrane and cytoplasmic concentrations. We agree that this may have been confusing given our demonstration that the mechanism that leads to cooperativity is dimerization-dependent reduction in membrane dissociation. In other words, it is dimerization-dependent stabilization of PAR-2 at the membrane that leads to the observed cooperativity.

However, we take issue with the implied assertions of lack of novelty. We do not feel a point-by-point comparison of our data and that of Arata et al is appropriate here - suffice it to say that the manuscripts have different scopes. Whereas Arata et al sought to understand PAR-2 polarity in the context of a full PAR reaction-diffusion model, which subsumed the potential contributions of PKC-3 activity and oligomerization into effective phenomenological feedback terms, we focus on understanding the isolated PAR-2 subcircuit and its molecular basis. In this regard, we would stress several points: First, the previous work did not identify the mechanism of dimerization or validate the effect of disrupting dimerization experimentally. Thus they did not demonstrate either that the RING drives dimerization or that dimerization contributes to polarization. Second, we measure and demonstrate (i) that membrane binding is cooperative, (ii) that cooperativity emerges from optimized dimer affinity, and (iii) that this optimized affinity is critical for proper function - none of these concepts are addressed in prior work.

2) Figure 1: I disagree with the first statement of the result section on how the binding of PAR-2 is regulated (mass action versus cooperativity). What happens if the cortical PAR-2 intensity is normalized to the length of the domain that it occupies? Also, if there is cooperativity in control, why the cooperativity is abolished in *par-3* mutants (where the domain occupied by PAR-2 is actually longer)?

Cooperativity is not abolished in *par-3* mutants. Figure 1G shows that the plots of membrane vs cytoplasmic concentrations for PAR-2 in WT and *par-3(it71)* overlay and fits of individual PAR-2 data for the two conditions yield apparent cooperativity near 2 in both cases as would be predicted if cooperativity is intrinsic to PAR-2 (Figure S2). The reduction in M:C ratio shown in *par-3(it71)* relative to wild-type (Figure 1E) is precisely what one predicts for concentration-dependent membrane binding. In *par-3(it71)*, PAR-2 is spread over a larger area resulting in reduced concentrations, which allows us to test whether this difference in concentration affects M:C ratios. Just to stress, if there was no concentration-dependence in membrane binding of PAR-2, M:C ratios should be the same for WT at the posterior and in *par-3(it7)* despite the difference in domain size. We show clearly that this is not the case.

In this same figure, the authors use the C56S mutant of PAR-2 but how this mutant influences membrane localization of PAR-2 is not clear. Is this mutant also interfering with PAR-2 dimerization? If yes, why the authors use other mutants later? How do the results compare to the L109R mutant (which is supposed to disrupt PAR-2 dimerization)? Again, as mentioned above, what are the data that support a feedback mechanism?

Figure 1G simply demonstrates that the C56S mutant exhibits reduced cooperativity, indicating that the RING domain is important. We then explore the mechanism of RING-dependent cooperativity in later figures. Regarding C56S vs L109R - as C56S would be expected to cause unfolding of the RING domain, we felt it “cleaner” to use a mutation that only targets the predicted dimer interface for the majority of the work in the manuscript. That said, the reviewer should note that data for equivalent assays for the two mutants were provided in Figure 1E, G and Figure 2I, J, which demonstrate that both mutants reduce cooperativity to a similar extent in vivo.

3) In the description of figure 2, the authors conclude: “Thus the RING domain appears to be sufficient to mediate recruitment by endogenous PAR-2 provided that it is stabilized at the plasma membrane”. I disagree with this conclusion: the domain alone cannot go to the plasma membrane at all. It can go only when fused to a membrane targeting domain, which means it is not sufficient.

We do not fully understand this criticism. We explicitly state in the text that the PAR-2 RING is not directly recruited by PAR-2 from the cytoplasm. However, if we localize the RING to the correct membrane compartment via an unrelated membrane binding domain that does not interact/colocalize with PAR-2, the RING can co-segregate with full length PAR-2 (i.e. segregating to the posterior in a RING- / PAR-2-dependent manner). In other words, the RING is indeed sufficient to direct a PM-associated molecule to co-segregate with PAR-2. We have clarified the text in the hopes it is now clear, specifically replacing “recruitment” with “co-segregation.”

4) In figure 2, the authors show dimerization of PAR-2 and the reduction of dimerization of the L109R. There is a major problem with this figure: in the text the authors say that the RING domain of PAR-2 shows a “concentration

dependent dimerization shifting from mostly monomeric to mostly dimeric". Now, one of the lowest concentrations used in the wild type PAR-2 RING domain is 0.75 mg/ml (Figure 2F) and this is mostly monomeric, as stated by the authors. To show that dimerization is lost in the L109R mutant and that this is monomeric, the authors used 0.75 mg/ml. At this concentration PAR-2 is already monomeric when wild type and indeed the difference compared to wild type in 2H is minimal. Why only the lowest concentration was used? The authors should use up to 10 mg/ml and show that dimerization is indeed affected in the mutant. In addition, are the values in the axis in figure 2G correct? As mentioned above, it would be interesting to test the C56S mutant in vitro and see if it influences dimerization.

We would disagree that the difference between WT and L109R is minimal at the concentration tested. We used the highest concentration available for the mutant and compared that directly with WT. At the tested concentration, WT is around ~35% dimeric, while L109R is only ~5% dimer (Figure 2F-H). We have now provided model fits showing that this difference equates to L109R having a roughly 10-fold reduction in dimer affinity (Figure 2H). Note this is roughly similar to the predicted reductions based on in vivo measurements (Figure 3F), which is now explicitly noted in the text. To make this point more clear, we have improved the presentation of these results, including a direct comparison of SEC MALS traces between WT and L109R at 750 ug/ml on similar axes, in which this difference is obvious (Figure 2G).

For C56S, despite multiple attempts, we were unable to purify usable quantities suggesting it is intrinsically unstable in vitro, consistent with the disruptive nature of this mutation to the RING domain architecture. This latter result further supports our choice to work primarily with the L109R mutation for the purpose of our dimerization analysis.

5) The total level of the different PAR-2 mutants/fusions needs to be measured and compared with the wild type, either by microscopy or by western blot. If the total level can be obtained by summing up the plasma membrane and cytoplasmic values in Table S2, this means that the mutants used (both C56S and L109R) both result in reduced total levels of PAR-2, which makes the interpretation of all the experiments harder and also is not consistent with the paragraph suggesting that the C56S mutant may have stronger phenotypes because of destabilization of the RING domain and lower protein amounts as the L109R seems to do the same.

Total concentrations for WT, L109R, and C56S were provided (see last column "Total Expression" in old Table S2, new Figure EV1). There are several points to unpack here.

- (1) While we obtain slightly different measures between WT and *par-3(it71)* conditions, L109R is generally present at higher levels than C56S. We also noted that C56S was unstable during purification. That is about all we can say.
- (2) While we do not fully understand precisely why C56S is worse, it is not clear to us why the referee is so concerned about the difference in C56S and L109R mutations. It is for precisely the reason that C56S is likely to disrupt folding of the RING domain and thus exhibit stability issues and potential pleiotropic effects that we introduced the L109R interface mutants upon which the rest of the paper is based. Understanding the precise reason why C56S shows stronger phenotypes is not really relevant to the core findings of the work.
- (3) The potential effects of protein dosage on phenotype is a salient issue, but the maximal reductions are less than 20%, a value that cannot explain the reduced membrane concentrations or reduced M:C ratios of RING mutants.

6) Statistics are missing in most if not all figures. As one example in figure 5H the authors say that the fraction of PAR-2(GCN4, PRBH) at the posterior PM is higher compared to PAR-2(GCN4) but the variability for one of the two is quite high and the difference in value is quite small. Is this significant? If the authors use the same scale in both 5F and H, PAR-2(GCN4) in 5H appears to have approximately the same value as the PAR-2(GCN4-PRBH) in 5H (a bit higher than 0.4), which is not consistent with the argument of the authors.

We have added a "Statistics" section to the Methods and provide a Supplemental Data file (Table EV3) detailing the comparisons and the relevant effect sizes and associated 95% confidence intervals.

For the specific example mentioned, we would point out that the key comparison is not between PAR-2(GCN4, PRBH) and PAR-2(GCN4). Rather, the relevant observation is the differential effect of the PRBH mutant in WT vs GCN4 backgrounds. PRBH mutations in an otherwise wild-type form of PAR-2 leads to a dramatic reduction in membrane binding (Δ M:C -0.347(0.373, 0.34)), while the same mutation in PAR-2(GCN4) leads to an increase (Δ M:C 0.0479(0.0281, 0.0684)). This difference is obvious from the data shown, but we have included the Δ M:C effect sizes and confidence intervals in Table EV3.

Note that axes in 5F and 5H are different given that in 5F we measure the total membrane pool (as we are comparing polarized and unpolarized conditions), while in 5H we only quantify within the posterior domain. Thus the numbers are not directly comparable. We have added a note to this effect to the figure legend.

7) In the plots, does each dot correspond to 1 embryo? It is hard to see how many embryos have been measured which should be indicated in the figure or figure legend together with the number of independent experiments performed.

Unless otherwise noted, all data points are shown for transparency. We have added "n=" where appropriate and noted this in the Statistics Table EV3.

8) The major concepts are often expressed in a convoluted manner which does not help the reader to understand. As a simple example, in page 7, the authors say: "Varying membrane binding.....". However, they actually vary the K(D), not the K(ON).

In the same sentence, the authors say: "for high membrane binding affinity" but they actually take a value that is in the middle (Fig. 3B). Why do they say "high", especially if compared to the value that they take for dimer affinity, which they call intermediate?

We have adjusted text throughout to address these points and standardized terminology.

9) To increase dimer affinity, the authors create a constitutive PAR-2 dimer by adding the GCN4 leucine zipper motif. One problem of this experiment is that PAR-2GCN4 goes to internal membranes. Additional controls are needed for this experiment. First, an endogenous PAR-2GCN4 fusion with no other tag would be a good control that the localization is indeed altered (in fixed samples by staining).

It is not necessarily clear to us why this is a necessary control. We already show that neither mNG::PAR-2 nor mNG::GCN4 go to these sites and have reproduced the effect with an independent dimerization domain. Given the effort required to generate the required lines, the added value seems minimal.

The open question is also: what are these internal membranes? And why only those? Why this dimer does not go, for example, to the ER or the Golgi? (assuming that these internal membranes are not ER or Golgi). So, the selectivity is not totally disrupted as the dimer does not seem to go to all internal membranes but only to a subset of those. What are these membranes? Colocalization with markers is needed to identify the membranes.

As noted above, we now show that PAR-2(GCN4) appears to localize generally to endosomal compartments, with the highest overlap with RAB-7 followed by RAB-5 (Figure EV3, Appendix Figures S3/S4). Given that PAR-2 is thought not to bind via specific lipid recognition, but via clusters of charged/hydrophobic residues, localization to endosomal compartments makes sense as these are thought to be the next most negatively charged relative to the PM and thus are likely to be the preferred membrane when the PM is unavailable due to PKC-3-mediated exclusion.

Have the authors considered the possibility that the reason why PAR-2(GCN4) is less at the plasma membrane is not because the dimerization strength is increased but because PAR-2 is unable to form larger oligomers that would be stabilized at the plasma membrane?

This is very unlikely. FRAP experiments in Illukkumbura et al (2023) already show that PAR-2(GCN4) turns over much more slowly than WT on the plasma membrane (we have now noted this in the text). The explanation proposed by the Referee also does not explain (1) why PAR-2(GCN4) localizes to internal membranes, (2) why cytoplasmic turnover of PAR-2(GCN4) is increased, or (3) why we can reduce internal accumulation by **reducing** membrane affinity. Their hypothesis would predict the opposite effect - i.e. that decreasing membrane affinity would lead to *more* internal accumulation of PAR-2(GCN4).

10) The fact that PAR-2(GCN4) in par-3 mutant embryos occupies less internal membranes suggests that this behavior of PAR-2(GCN4) (binding to internal membranes) is specific to embryos and not oocytes/germline. If this is the case, what is the explanation?

The key aspect of this experiment is not gonad vs. embryo, but rather the fact that normally PAR-2 is acutely removed from the plasma membrane by PKC-3 at the end of meiosis II. Thus PAR-2 must normally redistribute from internal pools to the posterior plasma membrane upon polarization. We show this occurs rapidly in WT, but is much

slower in PAR-2(GCN4) (Figure EV4). This acute removal at the end of meiosis does not occur in *par-3* mutants and hence the distribution of PAR-2 should more closely reflect the expected steady-state as PAR-2 has had more time to equilibrate between internal and plasma pools. As predicted by the model, increasing this time for equilibration via *par-3* mutation reduces the observed accumulation on internal membranes.

In Figure S8B, it seems that PAR-2(GCN4) remains longer at the plasma membrane (picture -20 minutes). Is this the case? And if yes, how do the authors reconcile with their hypothesis? It does not look like here PAR-2 is trapped already in internal membranes.

Yes, this observation is consistent with Figure 5E in which we see that PAR-2(GCN4) is somewhat less efficiently cleared from the anterior, presumably because phosphorylation must overcome the higher avidity of the constitutive dimer.

We would point out the difference between -30 and -20 timepoints in which membrane concentrations appear to drop slightly and one sees increased granularity in the cytoplasm of PAR-2(GCN4) compared to the near uniform distribution of PAR-2(WT), consistent with our model in which PAR-2 accumulates on these membranes when membrane concentrations are actively reduced by PKC-3 at the end of meiosis II.

Finally, in figure 5E, the PAR-2GCN4 domain is indeed less intense in wild type but extends more anteriorly.

This mutant is carried more efficiently by anterior-directed flows due to longer membrane residence times (see Illukkumbura et al, 2023) and is also somewhat more resistant to PKC-3 activity as described above. Together these features would explain why it tends to exhibit slightly extended domains.

Minor comments

In the abstracts the authors mention "ubiquitin independent function" of the RING domain. As there are no data about ubiquitin, the sentence should be removed.

As it is peripheral to the study, we have removed all references to ubiquitin-independence.

At page 3, in the second paragraph, the authors say: "...and which are thought to be targeted for phosphorylation...": why are they thought? Are the authors not convinced by the published literature?

We agree this is the most likely model, but we are unaware of any direct experimental validation of PKC-3 phosphorylation sites within PRBH domains. Note PAR-2(S241A), which blocks most PAR-2 phosphorylation by PKC-3, is not in a formally reported PRBH domain (Motegi et al 2011). However, given this concern we have simplified the text to say "its[PAR-2] dissociation from the membrane is promoted by phosphorylation by PKC-3".

The numbers in the PAR-2 scheme in Figure 1B are not clear as they do not correspond to the domains mentioned.

We have adjusted the scheme.

Can the authors show images of the L109R embryos?

Sample images have now been added - Appendix Figure S1E.

To reduce the flows, the authors use *nop-1* in one experiment and *mlc-4* in another. Any reason for this? Why not using *nop-1* in both cases?

These are different experiments. In Figure 2K, L we are looking at synthetic effects of reduced flow and dimerization mutants on division asymmetry and viability. *mlc-4(RNAi)* disrupts division in the zygote and so is unsuitable for this

purpose, hence the use of *nop-1*. In Figure 4, we are looking at robustness of PAR-2 domain formation in the no-flow condition and so use *m/c-4* to more fully compromise cortical flows.

In the scheme in figure 1H is practically impossible to see a reduced membrane PAR-2.

We have adjusted shading for clarity.

Describing the result, the authors state: "We found that PAR-2(GCN4) was enriched at the posterior as PAR-2(WT)". The statement is incorrect as the levels of the PAR-2 fusion at the posterior are much lower than the WT, as stated just below in the text.

We simply meant that it was posteriorly localized (like WT is). Indeed we explicitly commented on the difference in levels in the next sentence. We have rephrased to make this clear.

Figure 4: for consistency, the authors should show the MC measurements as in figure 1 and 2.

We chose this measure deliberately. Given that the true cytoplasmic signal is obscured by internal membrane binding, the M:C ratio would not be meaningful or comparable to Figure 1 and 2 in which there is no visible internal membrane binding.

It would be useful if the authors put in the PAR-2 scheme where the GCN4 domain is introduced

We state in the text that this is introduced immediately after the RING domain and felt it would likely be confusing to readers to introduce the GCN4 domain already in Figure 1.

When the authors introduce the experiment where they mutate the serines in the PRBH domain, it would be good if they add a sentence introducing the rationale of the experiment.

We have altered the text to improve clarity.

Referee #3:

Bland et al. present very interesting research addressing a long-standing question in the field of PAR polarity, how PAR-2 pathway can induce zygote polarization (symmetry breaking) in the absence of the actomyosin flow-induced pathway. The problem had been formulated a while ago, mostly in the pioneering works from the lab of Geraldine Seydoux, which are duly cited. In those papers, it had been shown that self-interaction of PAR-2 via the RING domain plays an important role. Here, the authors further develop this theme and show the crucial contribution of the RING domain dimerization. From the point of view of biological experiment, the manuscript presents careful, meticulous analysis with thoughtful controls and excellent image analysis. I would like to commend the authors on the sheer volume of work, creative use of mutants and attention to detail. This work leads the authors to conclude that the ring domain dimerization introduces positive cooperativity into the binding of PAR-2 to the membrane.

My major issue is with the presented theoretical explanation. The authors take a highly non-standard approach, developing a novel thermodynamics-based theory, completely avoiding the use of routine reaction-diffusion formalism. The biological question is how PAR-2 subsystem can explain cellular polarization - i.e., how this mechanism can induce symmetry breaking. However, this very central question is not addressed by the proposed theory! The authors must provide such an analysis and clearly demonstrate that their simple mechanism can, indeed, break symmetry, e.g., by using linear stability analysis of the homogeneous state of the cell. If the authors claim that this is the case, then we have another problem of fundamental physics nature. In this case, the claim is that any protein that can reversibly dimerize and bind to the membrane is sufficient to break spatial symmetry! This is too big a claim and, frankly, sounds too good to be true. As the system consists of only reversible reactions without any input of energy, i.e., purely equilibrium system, then how does it break symmetry? This seems to contradict the results typically associated with the seminal work of Ilya Prigogine. The authors have to squarely address this puzzling conundrum in both results and discussion and provide clear evidence and analysis.

The referee appears to have two main concerns:

- (1) The choice of model framework. We explicitly chose a thermodynamic approach as we wanted to assess how membrane binding and dimerization could account for the observed cooperative behavior without invoking an active, energy-driven process as this occurs in the absence of phosphorylation by PKC-3. Reaction-diffusion models are not necessarily thermodynamically consistent in this regard and can prove complicated for use in discerning thermodynamic aspects like binding affinities and dimerization strengths. We cannot exclude a role for energy input, but dimerization and membrane binding appear fully sufficient to account for the data and nicely match experimental data where it is available.
- (2) Whether our model captures symmetry-breaking. As noted above, our model was intended to capture PAR-2 membrane binding, not to model polarization. As the referee notes, given that it is an equilibrium model, it cannot polarize (or break symmetry) by construction as there is no energy input. We apologize if we gave the impression otherwise. The lack of polarization by this isolated system actually makes sense given that PAR-2 requires PKC-3 activity (though not PKC-3 asymmetry) to polarize in all known contexts, including the no-flow regime where the PAR-2 pathway is required. Thus, explaining polarization would require integrating this dimerization-dependent cooperative membrane binding behavior of PAR-2 within a larger network that takes into account PKC-3 activity, which is beyond the scope of the current work.

Referee #3 comments on the revision plan:

I would like to thank the authors for their detailed reply to my criticism. I do appreciate that they do not clearly claim that they provide a mechanism for "symmetry breaking". However, they do use word "polarization" frequently in the body in conjunction with the PAR2 domain formation. The term "polarization" is for many people equivalent to "symmetry breaking". I strongly recommend using instead "PAR2 domain formation" or equivalent to prevent any ambiguity. Further to my early suggestions and taking into account the authors' reply, here are my recommendations:

1. 1. As the lead author is known for his contribution to the "flow pathway", while current work mainly concerns with the "PAR2 pathway" with which other labs are closely associated, in the Introduction, it would be very useful to add a paragraph succinctly recounting the event sequence happening during the zygote polarization. Too many people know something about PAR polarity complexes but very few have a clear view of normal and "alternative" pathways of *C. elegans* zygote polarization. It would be beneficial to highlight that the primary symmetry-breaking event is the entry of the sperm, causative of the local cortex weakening and the following "flow pathway". Offering the readers a few modern reviews, e.g., Kapoor and Kotak PubMed 32597472, would be helpful.
2. 2. In the Discussion, the authors should clearly indicate where exactly in the sequence of events throughout normal and "alternative" polarity establishment they see the role of PAR2 binding cooperativity. This is greatly important to put the results of the paper in the perspective and avoid any potential misunderstanding

In light of these comments, we have attempted to be much clearer throughout in our notation and have included additional details regarding flow-dependent and PAR-2-dependent pathways for polarization (see Introduction and revised Figure 1A). We have also summarized these points explicitly in the Discussion to place our work in the proper context.

To reiterate, our data suggest that oligomerization is dispensable for PAR polarity when cortical flows are the primary symmetry-breaking event - in such cases, pPARs largely serve to reinforce the initial flow-induced asymmetry of aPARs. By contrast, cooperativity becomes **necessary** for polarity when embryos must polarize via the PAR-2 pathway and PAR-2 domain formation is the first sign of polarity. However, we emphasize that cooperativity is **not sufficient** for PAR-2 domain formation in these conditions - rather domain formation still requires PKC-3 activity and hence energy input (though PKC-3 need not be, itself, segregated). Hence to model polarization in the no-flow regime, one must include the energy-dependent interaction between aPARs and pPARs such as phosphorylation of pPARs by PKC-3.

Dear Nate,

Thank you for submitting a revised version of your manuscript. Your study has now been seen by all original referees, and you can find their comments below. While reviewers #1 and #3 find that most of their previous concerns have been addressed, reviewer #2 raises several points that would still need to be addressed in the final minor revision. I have consulted with reviewer #1 on these remaining points, and based on their additional input, please address these concerns as follows:

- 1) Point 1 on statistics - please add a clarification and explanation for the used bootstrapping in the Materials and Methods section, and include the information on the used statistical tests in the figure legends.
- 2) In response to point 2 (original point 6), please refer to Figure 5E in the following sentence to guide the readers to the relevant panel:
"Consistent with reduced kinetic trapping, we found that PAR-2(PRBH, GCN4) exhibited reduced accumulation on internal membranes compared to PAR-2(GCN4) (Figure 5E)"
- 3) Point 3 (original point 5) - please add further discussion on the potential effects of PAR-2 mutant transgene expression levels on the observed phenotypes.

Please also address textually the remaining comment by reviewer #1.

There also are a few editorial points that need to be sorted out before I can extend formal acceptance of the manuscript:

1. Please reduce keywords to maximum five.
2. Please check the email address for Neil Q McDonald, the emails sent to this address could not reach the addressee.
3. Please remove OrCID numbers from the title page.
4. Please check that the funding information is correct and identical both in the manuscript and our online system. NIH Office of Research Infrastructure Programs (P40 OD010440) is currently missing in our online system.
5. CRediT has replaced the traditional author contributions section because it offers a systematic, machine-readable author contributions format that allows for more effective research assessment. Please remove the Authors Contributions from the manuscript and use the free text boxes beneath each contributing author's name in our online submission system to add specific details on the author's contribution. More information is available in our guide to authors.
6. Please rename "Conflict of interest" section into "Disclosure and competing interests statement" (further info: <https://www.embopress.org/page/journal/14602075/authorguide#conflictsofinterest>).
7. During our standard figure check, we noticed that figure panels 2A and 4C mNG appear to be derived from the same image. Please either correct or add a sentence in the figure legend to indicate the reuse of the same image.
8. Figure panel 1H is not mentioned in the manuscript text.
9. Please upload the tables in an Excel format and add the table legends to the file.
10. In the Appendix, please update the nomenclature to Appendix Figure S1, etc.
11. Please remove the list of Supporting Information from the manuscript text file.
12. Majority of the submitted source data files for Fig 1D, 2A-B, 2I-K, 4A (wt), 4B-E, 5E-F, 5G (PRBH) appear empty (zero bytes), please check - there might have been a problem during the upload.
13. Our data editors have flagged the following issues in figure legends that need correcting:
 - Please add that information on the number and nature of replicates in the legends of figures 5f, h; EV1a, d, f.
 - Please define the measure of centre for the error bars in the legends of figures 1f-g; 2j; EV1d, f.
 - Please add that scale bar and its definition for figure EV4a.

With best wishes,

leva

leva Gailite, PhD
Senior Scientific Editor
The EMBO Journal
Meyerhofstrasse 1
D-69117 Heidelberg
Tel: +4962218891309
i.gailite@embojournal.org

We realize that it is difficult to revise to a specific deadline. In the interest of protecting the conceptual advance provided by the work, we recommend a revision within 3 months (20th Jun 2024). Please discuss the revision progress ahead of this time with the editor if you require more time to complete the revisions.

Referee #1:

Bland et al. have satisfactorily addressed all the issues I have previously raised. I appreciate the authors' efforts to improve the manuscript.

My only concern with the revised manuscript is the vague definition of 'responsiveness' in terms of a role for PAR-2 in establishing polarity. I understand that the authors have provided evidence for oligomerisation of PAR-2 in the robustness of PAR polarity establishment, as loss and excessive oligomerisation impaired asymmetric cell division in a particular genetic background, *nop-1*(RNAi). However, the phenotypes of PAR-2 oligomerisation mutants in this manuscript do not provide insight into the 'responsiveness' of PAR polarity. How the PAR-2 polarity system can adapt to changes in the cellular state during development is completely unclear. No results in this manuscript support a role for PAR-2 oligomerisation in sensing and adopting the symmetry-breaking cue. Therefore, I strongly suggest that the authors tone down and revise the claim and discussion of 'responsiveness' in the manuscript.

Referee #2:

This revised version of Bland et al has definitely improved and some parts are more clear and easier to read. However, there are still important comments that have not been addressed and need to be addressed before publication.

Both referee 1 and referee 2 ask for statistical comparison in many of the figures. Instead of providing statistics in each figure and clearly indicating what is significant and what not (in the figure itself or the figure legend), the authors provide a table in the extended information. In addition, the table does not directly compare data and their significance, but the authors use bootstrapping and provide the CI for the data. Why such a convoluted manner? Why using bootstrapping? Please provide in the main text for each quantification the appropriate statistics and the p value. Referee 1 also asks, rightly, for the number of experiments performed. The authors only report the number of embryos analyzed but not the number of experiments. Does it mean that experiments were done only once? I suppose not.

In their response to point 5, the authors say that they think it is unlikely that a 20% difference in protein levels (in the case of the C56S mutant) can account for the observed phenotype in cooperativity. This is of course a speculation as they do not know. When they explain the difference in terms of lethality and phenotype between the C56S and the L109R mutants, the authors say that this pretty strong difference in phenotype (lethality versus viability) could be due to the difference in protein levels of these two mutants. However, in their table, the two mutants differ way less than 20% between each other. So, how can a small difference in protein levels explain a strong difference in one phenotype (lethality versus viability) but a bigger difference cannot explain the cooperativity phenotype?

For point 6 of my original review (comparison between the GCN4 fusion and the GCN4-PRBH) I disagree with the authors. In their rebuttal, the authors say: the key comparison is not between PAR-2(GCN4, PRBH) and PAR-2(GCN4) rather between PAR-2PRBH and PAR-2(GCN4, PRBH). First, I believe this comparison is important since the authors want to show "rescue" of kinetic trapping (which occurs in the GCN4 fusion). Second, the authors indeed keep comparing these two situations in the text (page 11, blue part). Their statement is:

Consistent with reduced kinetic trapping, we found that PAR-2(PRBH, GCN4) exhibited reduced accumulation on internal membranes compared to PAR-2(GCN4)(Figure 5G)

but, in 5G, PAR-2(GCN4) is not even shown.....so, how can the reader compare?

Just below the authors say:

not only does PAR-2(PRBH, GCN4) not exhibit reduced accumulation relative to PAR-2(GCN4), but actually accumulated to slightly higher levels (Figure 5H).

again comparing the two PAR-2 forms that, in the rebuttal letter, they say are not important to compare.

Minor comments

Page 5: I find the sentence "which features of PAR-2 were responsible for these polarization dependent changes in membrane binding" misleading (and probably referee 3 as well. It makes it sound like the cooperativity occurs only in polarized embryos (which is indeed not the case). What about "concentration dependent"?

Page 11: I am not sure that saying that a par-3 mutant gives "more time" for equilibration is correct. Since par-3 is not there, PAR-2 can localize from the start at the cortex. I would rephrase. The question is: why at this stage there is none that goes to the internal membrane (not that this question should be addressed)?

In figure 4D and E it would be good to also have the PAR-2(GCN4) control (no mlc-4(RNAi))

The author should discuss why, if ring dimerization is important, the PAR-2(178-412) which does not contain the RING domain (Hao et al), can enrich at the cortex.

Total expression of PAR-2 (GCN4) and PAR-2(GCN4PRBH) are missing in the table

Referee #3:

I am satisfied with the revisions provided by the authors and recommend publication of the manuscript.

Response to Referee Reports (1 May 2024)

Thank you for your careful evaluation of our manuscript and additional suggestions for improvement. We have now addressed all concerns raised as detailed below. Please note that we largely disagree with the primary concerns of Referee 2 as laid out in our detailed response below. Changes to the text are highlighted in blue in our manuscript.

Thank you for submitting a revised version of your manuscript. Your study has now been seen by all original referees, and you can find their comments below. While reviewers #1 and #3 find that most of their previous concerns have been addressed, reviewer #2 raises several points that would still need to be addressed in the final minor revision. I have consulted with reviewer #1 on these remaining points, and based on their additional input, please address these concerns as follows:

1) Point 1 on statistics - please add a clarification and explanation for the used bootstrapping in the Materials and Methods section, and include the information on the used statistical tests in the figure legends.

We have added additional notes to the Methods, included a suitable reference, and now note the use of bootstrapping more extensively in the Figure legends for clarity.

2) In response to point 2 (original point 6), please refer to Figure 5E in the following sentence to guide the readers to the relevant panel:

"Consistent with reduced kinetic trapping, we found that PAR-2(PRBH, GCN4) exhibited reduced accumulation on internal membranes compared to PAR-2(GCN4) (Figure 5E)"

We have added additional signposting to this data.

3) Point 3 (original point 5) - please add further discussion on the potential effects of PAR-2 mutant transgene expression levels on the observed phenotypes.

We have added additional comments on interpreting the C56S allele phenotypes and now show that we do not observe such phenotypes in *par-2* heterozygotes (Appendix Figure S1F).

Please also address textually the remaining comment by reviewer #1.

See response below.

There also are a few editorial points that need to be sorted out before I can extend formal acceptance of the manuscript:

1. Please reduce keywords to maximum five. Done
2. Please check the email address for Neil Q McDonald, the emails sent to this address could not reach the addressee. Done
3. Please remove OrcID numbers from the title page. Done
4. Please check that the funding information is correct and identical both in the manuscript and our online system. NIH Office of Research Infrastructure Programs (P40 OD010440) is currently missing in our online system. Done
5. CRediT has replaced the traditional author contributions section because it offers a systematic, machine-readable author contributions format that allows for more effective research assessment. Please remove the Authors Contributions from the manuscript and use the free text boxes beneath each contributing author's name in our online submission system to add specific details on the author's contribution. More information is available in our guide to authors. Done
6. Please rename "Conflict of interest" section into "Disclosure and competing interests statement" (further info: <https://www.embopress.org/page/journal/14602075/authorguide#conflictofinterest>). Done
7. During our standard figure check, we noticed that figure panels 2A and 4C mNG appear to be derived from the same image. Please either correct or add a sentence in the figure legend to indicate the reuse of the same image. We have now included an alternative image.
8. Figure panel 1H is not mentioned in the manuscript text. This is now included.

9. Please upload the tables in an Excel format and add the table legends to the file. Done
 10. In the Appendix, please update the nomenclature to Appendix Figure S1, etc. Done
 11. Please remove the list of Supporting Information from the manuscript text file. Done
 12. Majority of the submitted source data files for Fig 1D, 2A-B, 2I-K, 4A (wt), 4B-E, 5E-F, 5G (PRBH) appear empty (zero bytes), please check - there might have been a problem during the upload. Done
 13. Our data editors have flagged the following issues in figure legends that need correcting:
 - Please add that information on the number and nature of replicates in the legends of figures 5f, h; EV1a, d, f. Done. Note that EV1d, f are probability distributions calculated by bootstraps - see methods. We have included additional language to make this clear.
 - Please define the measure of centre for the error bars in the legends of figures 1f-g; 2j; EV1d, f. These are all best fits to the full dataset. Additional language added to Figure legends.
 - Please add that scale bar and its definition for figure EV4a. Done
-

Referee #1:

Bland et al. have satisfactorily addressed all the issues I have previously raised. I appreciate the authors' efforts to improve the manuscript.

We thank the Referee for their constructive comments/criticism.

My only concern with the revised manuscript is the vague definition of 'responsiveness' in terms of a role for PAR-2 in establishing polarity. I understand that the authors have provided evidence for oligomerisation of PAR-2 in the robustness of PAR polarity establishment, as loss and excessive oligomerisation impaired asymmetric cell division in a particular genetic background, *nop-1(RNAi)*. However, the phenotypes of PAR-2 oligomerisation mutants in this manuscript do not provide insight into the 'responsiveness' of PAR polarity. How the PAR-2 polarity system can adapt to changes in the cellular state during development is completely unclear. No results in this manuscript support a role for PAR-2 oligomerisation in sensing and adopting the symmetry-breaking cue. Therefore, I strongly suggest that the authors tone down and revise the claim and discussion of 'responsiveness' in the manuscript.

Nothing has changed in the revision with respect to our discussion of "responsiveness" and it would have been helpful if the reviewer could have clarified in their initial comments what they would have liked to see here. While we agree that "responsiveness" is a somewhat vague term, a key point we make is that if dimer affinities are too strong, PAR-2 binds too tightly to membranes and thus redistributes too slowly from internal membranes back to the plasma membrane during polarization. Specifically, constitutively dimeric PAR-2 shows reduced loading onto the posterior membrane in response to aPAR clearance by cortical flows (Figure 4A) and reduced ability to drive polarity in response to suboptimal symmetry-breaking cues (no flow, Figure 4D). Thus, optimized dimer affinity is required for the optimal responsiveness of PAR-2 to the changes in signals that occur at the meiosis-mitosis transition in which PAR-2 shifts from being excluded globally from the PM to being locally loaded in the posterior PM, with consequences for the efficiency of polarity, at least in suboptimal symmetry-breaking conditions. Perhaps the Reviewer has something different in mind with respect to "responsiveness," but from the comments it is unclear what their criteria for determining whether a system is more/less responsive is and what evidence would be required in this regard. We have nonetheless gone through and slightly adjusted text to make our points more clear.

Referee #2:

This revised version of Bland et al has definitely improved and some parts are more clear and easier to read. However, there are still important comments that have not been addressed and need to be addressed before publication.

Both referee 1 and referee 2 ask for statistical comparison in many of the figures. Instead of providing statistics in each figure and clearly indicating what is significant and what not (in the figure itself or the figure legend), the authors provide a table in the extended information. In addition, the table does not directly compare data and their significance, but the authors use bootstrapping and provide the CI for the data. Why such a convoluted manner? Why using bootstrapping? Please provide in the main text for each quantification the appropriate statistics and the p value. Referee 1 also asks, rightly, for the number of experiments performed. The authors only report the number of embryos analyzed but not the number of experiments. Does it mean that experiments were done only once? I suppose not.

The use of simplistic, binary significance reporting via p values is increasingly seen as problematic, not least because they typically provide no information of effect size. There are countless reports of statistically significant, but biologically irrelevant differences in the literature.

Here we have provided our data in a fully transparent manner, showing all data points where possible, which allows readers to judge the data for themselves and in most cases, the effects are obvious. That said, unlike what the reviewer implies, in response to prior comments, **we provide direct effect size analysis with confidence intervals for the relevant comparisons, from which the reader can infer statistical significance from the lack of overlap in 95% confidence intervals obtained by bootstrap analysis.**

Bootstrapping is a rigorous, well-established statistical method that often performs better than standard t-tests. It relies on fewer assumptions regarding the underlying data, and the effect size analysis we provide is more transparent in providing information both on the significance AND magnitude of the differences. If the reviewer had a scientific argument for why bootstrapping and effect size analysis are inappropriate, they could have provided this. They did not.

Our manuscript is not the appropriate venue for a discussion of the pros and cons of bootstrapping, confidence intervals, and p values and we instead refer the reviewer and readers to a variety of recent reviews/commentaries on the subject:

<https://www.scientificamerican.com/article/the-significant-problem-of-p-values/>
<https://www.tandfonline.com/doi/full/10.1080/00031305.2019.1583913>
<https://thenode.biologists.com/p-value-parroting/>
<https://thenode.biologists.com/a-better-bar/>

The choice to report statistics in the supplement allows us to maintain maximal visibility of data in the Figure panels so that readers can visually assess the data themselves. It does require slightly more effort for the reader to assess “significance” where they feel it is necessary, but we feel the tradeoff for increased data visibility and avoidance of overly simplistic assessment of significance is reasonable.

In their response to point 5, the authors say that they think it is unlikely that a 20% difference in protein levels (in the case of the C56S mutant) can account for the observed phenotype in cooperativity. This is of course a speculation as they do not know. When they explain the difference in terms of lethality and phenotype between the C56S and the L109R mutants, the authors say that this pretty strong difference in phenotype (lethality versus viability) could be due to the difference in protein levels of these two mutants. However, in their table, the two mutants differ way less than 20% between each other. So, how can a small difference in protein levels explain a strong difference in one phenotype (lethality versus viability) but a bigger difference cannot explain the cooperativity phenotype?

These comments misrepresent our claims and are factually incorrect. Here is our text from our previous response:

“The potential effect of protein dosage on phenotype is a salient issue, but the maximal reductions are less than 20%, a value that cannot explain the reduced membrane concentrations or reduced M:C ratios of RING mutants.”

First, note that this is very different from what the Reviewer claims we state in their comment (“it is unlikely that a 20% difference in protein levels can account for the observed phenotype in cooperativity”). Cooperativity is the

concentration-dependent change in membrane binding. Thus, **it makes no sense to say that reduced protein concentration would affect cooperativity**. A 20% reduction in the WT protein concentration would simply reflect a different point on the WT curve. The key point is that the C56S and L109R curves do not overlay with WT and so concentration differences alone do not explain the differences in membrane binding behavior.

Second, we state in our response to Reviewer 2's concerns that the reduction in **membrane concentrations and M:C ratios** between WT and C56S are unlikely to be explained solely by a 20% dosage reduction. **This is NOT simply speculation as Reviewer 2 suggests - we can say this precisely because we know exactly what the M:C ratio is for the WT protein at 80% of normal levels because this is what we measure in our RNAi experiments that underlie Figure 1G**. If the Reviewer would prefer to see data plotted in a more traditional fashion, please see Rodrigues et al *bioRxiv* 2023 - Figure 4H, where, using different lines and a slightly different quantification regime, we see that a ~20% reduction in PAR-2 amounts leads to a ~25% reduction in membrane concentration. Thus, the change in dosage cannot explain the >75% reduction in membrane accumulation in RING mutants relative to WT.

Third, the Reviewer again misrepresents our statements, implying that we argue that the difference between C56S and L109R is due to dosage. In fact, in our prior response we state:

"we do not fully understand precisely why C56S is worse."

We do not make any strong claims about why C56S exhibits a stronger phenotype mostly because it is largely irrelevant to the core message of the paper. We could remove all C56S data from the manuscript and it would not alter any of our conclusions. We present the C56S data as it was how we identified a role for the RING domain in cooperativity, which we then validated with the more selective L109R mutation. The most important observations are that both C56S and L109R show similar reductions in cooperativity. We do note that C56S exhibits a somewhat stronger lethality phenotype, which is expected given that C56S will destabilize folding of the RING domain, compared to L109R which is predicted to sit at the surface and only modify the dimer interface. We also mention that the modestly reduced levels of C56S relative to L109R are reasonably consistent with the expected destabilizing effect of C56S - that is all. As we stressed in our prior response, we don't understand the reviewer's concern regarding the differences between C56S and L109R as they are not relevant to the core findings of the paper and the Reviewer provides no answer to this point.

Finally, to eliminate any doubt that protein dosage could account for the observed phenotypes in RING mutants, which was already unlikely given the very modest reductions in expression (max ~20% lower), we show that there is no synthetic lethality of *nop-1(RNAi)* with *par-2(ok1723)/+* heterozygotes, which we know exhibit an ~40% reduction in PAR-2 amounts (Appendix Figure S1F)(<https://doi.org/10.1101/2023.11.21.568006>).

For point 6 of my original review (comparison between the GCN4 fusion and the GCN4-PRBH) I disagree with the authors. In their rebuttal, the authors say: the key comparison is not between PAR-2(GCN4, PRBH) and PAR-2(GCN4) rather between PAR-2PRBH and PAR-2(GCN4, PRBH). First, I believe this comparison is important since the authors want to show "rescue" of kinetic trapping (which occurs in the GCN4 fusion). Second, the author indeed keep comparing these two situation in the text (page 11, blue part). Their statement is:

Consistent with reduced kinetic trapping, we found that PAR-2(PRBH, GCN4) exhibited reduced accumulation on internal membranes compared to PAR-2(GCN4)(Figure 5G)

but, in 5G, PAR-2(GCN4) is not even shown.....so, how can the reader compare? *(Note that this data was present in Figure 5E).*

Just below the author say:

not only does PAR-2(PRBH, GCN4) not exhibit reduced accumulation relative to PAR-2(GCN4), but actually accumulated to slightly higher levels (Figure5H).

again comparing the two PAR-2 forms that, in the rebuttal letter, they say are not important to compare.

Unfortunately, it appears that the reviewer still fails to grasp the core points of this experiment.

First, when comparing kinetic trapping of PAR-2(GCN4) vs PAR-2(GCN4, PRBH), the key results are not in Figure 5H, but shown in the comparison between the PAR-2(GCN4) image in Figure 5E (and 4A for that matter) and the corresponding image of PAR-2(GCN4, PRBH) in 5G, as well as the FRAP experiments in Figure EV4. It is clear from these data that the PRBH mutation reduces accumulation on internal membranes and restores wild-type cytoplasmic turnover times, consistent with reduced kinetic trapping, which is exactly what we say in the text. We have added signposting to 5E as it appears the reviewer overlooked this panel.

Second, in Figure 5H, we compare plasma membrane accumulation (i.e. fraction at posterior membrane) of the various alleles. Here, the key observation is that the effect of PRBH mutations is different in the WT and GCN4 backgrounds. Specifically, these comparisons allow us to show, first, that PRBH mutations dramatically reduce membrane accumulation of PAR-2(WT), indicating that the mutation reduces the membrane binding affinity of PAR-2 as would be expected from adding S>E mutations in the BH domain. However, the comparisons additionally show that adding this mutation to the GCN4 allele does not reduce membrane binding (it actually seems to go up by a bit, but this isn't really important - the key is that it does not go down as much as it does in WT). This very non-intuitive result that reducing membrane binding affinity does not reduce plasma membrane accumulation in the GCN4 context is exactly what one would predict from the kinetic trapping model - the reduced membrane binding affinity is balanced out by the reduced kinetic trapping.

While the difference between PAR-2(GCN4) and PAR-2(GCN4, PRBH) in Figure 5H is significant (which can be ascertained from the non-overlapping 95% CI of the means that was added in the Supplement at the Reviewer's request), this is a good example of where a simple report of significance is not informative and simply reporting p values for all the pairwise comparisons is not useful. Take the hypothetical situation in which PAR-2(GCN4) and PAR-2(GCN4, PRBH) were similar. The key observation would be the same - the effect of the PRBH mutation is dramatically and obviously different in the WT and GCN4 contexts - In WT, PRBH reduces plasma membrane accumulation dramatically. In GCN4, it does not. Whether it is the same or goes up a bit is largely irrelevant.

Minor comments

Page 5: I find the sentence "which features of PAR-2 were responsible for these polarization dependent changes in membrane binding" misleading (and probably referee 3 as well. It makes it sound like the cooperativity occurs only in polarized embryos (which is indeed not the case). What about "concentration dependent"?

We have rephrased to avoid confusion. We are glad that Reviewer 2 now agrees with us that cooperativity is indeed not changed between WT and unpolarized conditions.

Page 11: I am not sure that saying that a par-3 mutant gives "more time" for equilibration is correct. Since par-3 is not there, PAR-2 can localize from the start at the cortex. I would rephrase. The question is: why at this stage there is none that goes to the internal membrane (not that this question should be addressed)?

The main point here is that because PAR-2 is not actively removed at the end of meiosis II in par-3 mutant embryos, the system should be closer to equilibrium. Whether it starts on the membrane or loads earlier is largely irrelevant. That said, in Reich et al (2019), we showed that PAR-2 is still cleared from the plasma membrane at ovulation in pkc-3(RNAi) embryos. It then gradually loads during meiosis, but fails to be removed at the end of meiosis II. Thus saying that the system has more time for PAR-2 membrane binding to reach equilibrium seemed more accurate.

In figure 4D and E it would be good to also have the PAR-2(GCN4) control (no mlc-4(RNAi))

A similarly-staged PAR-2(GCN4) embryo just before NEBD is shown in Figure 4A. These embryos are already fully polarized by the end of establishment phase, so there is little change for the relevant timeframe shown in 4E and so showing additional timepoints for this period would add little value.

The author should discuss why, if ring dimerization is important, the PAR-2(178-412) which does not contain the RING domain (Hao et al), can enrich at the cortex.

PAR-2(178-412) binds the membrane for the same reason as RING mutants do - they all contain the intact membrane targeting domain defined in Hau et al. The RING mutants just bind less stably as they lack dimerization-dependent stabilization. We see no reason to comment additionally on this allele, particularly given it was only analyzed as an ectopic transgene and there is a lack of quantitative analysis of either M:C ratios or expression levels for 178-412 with which to compare to our data. These caveats aside, Figure 2D from Hau et al appears to show that PAR-2(178-412) enrichment at the posterior membrane is reduced compared to PAR-2(WT), consistent with our results. Note also that although Hau et al do not report on PAR-2(178-412) behavior in the absence of endogenous wild-type PAR-2, they do show PAR-2(178-628), which also lacks the RING domain, and its behavior in embryos lacking wild-type PAR-2 is similar to the two RING mutants (C56S, 7C/S). Thus data from Hau et al are fully compatible with our conclusions and our model explains their data.

Total expression of PAR-2 (GCN4) and PAR-2(GCN4PRBH) are missing in the table

We have now added the relevant concentration data for all alleles, which are consolidated in Table EV1.

Referee #3:

I am satisfied with the revisions provided by the authors and recommend publication of the manuscript.

We thank the Referee for their constructive comments/criticism.

Dear Nate,

Thank you for addressing most of the final editorial points. I am now pleased to inform you that your manuscript has been accepted for publication. Congratulations on a nice study!

Before we forward your manuscript to our publishers, there are a couple of points that need to be addressed:

1) The title differs between the online submission and the manuscript. I find the version in the manuscript text more broadly accessible, and would like to propose a minor edit.

Title:

Optimized PAR-2 RING dimerization mediates cooperative and selective membrane binding for robust cell polarity

2) I would also like to suggest some minor edits in the manuscript abstract and synopsis (please also see the attached file). I have also written a short blurb that will accompany the title of your manuscript in our online system. Please let me know if any corrections are needed.

Blurb:

Dimerization properties of the posterior polarity protein PAR-2 regulate its stability at the plasma membrane to maintain polarization of the *C. elegans* zygote.

Synopsis:

Establishment of cell polarity is mediated by PAR protein pattern formation and is thought to rely on complex feedback responses. This study identifies membrane-dependent dimerization of the posterior polarity protein PAR-2 via its RING domain as one source of such feedback, which drives cooperative and selective stabilization of PAR-2 at the plasma membrane.

- PAR-2 exhibits cooperative plasma membrane binding.
- Cooperativity emerges from concentration-dependent PAR-2 RING domain dimerization.
- Increasing or decreasing dimer affinity compromises robustness of polarization.
- Optimized dimer affinity allows rapid and selective plasma membrane targeting.

If you have any questions, please do not hesitate to contact the Editorial Office. Thank you for this contribution to The EMBO Journal and congratulations on a successful publication!

With best wishes,

leva

leva Gailite, PhD
Senior Scientific Editor
The EMBO Journal
Meyerohofstrasse 1
D-69117 Heidelberg
Tel: +4962218891309
i.gailite@embojournal.org
